# Uncovering developmental time and tempo using deep learning

Nikan Toulany [1,2,3,5], Hernán Morales-Navarrete [1,4,5], Daniel Čapek [1], Jannis Grathwohl[1], Murat Ünalan [1,2,6] & Patrick Müller [1,2,3,4,6]

During animal development, embryos undergo complex morphological changes over time. Differences in developmental tempo between species are emerging as principal drivers of evolutionary novelty, but accurate description of these processes is very challenging. To address this challenge, we present here an automated and unbiased deep learning approach to analyze the similarity between embryos of different timepoints. Calculation of similarities across stages resulted in complex phenotypic fingerprints, which carry characteristic information about developmental time and tempo. Using this approach, we were able to accurately stage embryos, quantitatively determine temperature-dependent developmental tempo, detect naturally occurring and induced changes in the developmental progression of individual embryos, and derive staging atlases for several species de novo in an unsupervised manner. Our approach allows us to quantify developmental time and tempo objectively and provides a standardized way to analyze early embryogenesis.

The development of an animal from a fertilized egg to a mature adult is a complex and multifaceted process that stereotypically and almost invariably produces body plans with species-specific features and appearance. During early embryogenesis, animals pass through similar and characteristic stages of development[1–3]. First, during cleavage and blastula stages, embryos produce the building blocks of the future body plan through a series of cell divisions. Second, during gastrula stage, the cells are specified and arranged to set up the initial body axes of the animal. Third, during organogenesis stages, cells are rearranged to form specialized tissue systems. Fourth, during segmentation stages, the tissue systems are subdivided into repeated parts along the anterior–posterior axis. Finally, during larval stages, the body is functionalized to form an autonomous and integrated feeding, moving, sensing and responding entity[1,4–14].

Our knowledge of these different developmental stages and the transitions between them has been derived from careful—but tedious—manual microscopic observation (Supplementary Note 1)[4–12]. Idealized images in the resulting species-specific atlases capture the essence of characteristic stages and link them to absolute developmental time, assuming that morphological traits are constant within a developmental stage and that stages can be correlated reliably with absolute measured time. However, in reality embryos rarely look like the idealized illustrations in staging atlases (Supplementary Fig. 1a), and transitions between developmental stages usually do not occur abruptly but smoothly (Supplementary Fig. 1b and Supplementary Video 1). The appearance of different phenotypic traits during development and the persistence of these traits over different lengths of time results in overlapping morphologies (Supplementary Fig. 1c and Supplementary Video 1), and it can therefore be difficult to strictly define sharp boundaries between subsequent developmental stages. Even when examining a group of sibling embryos at the same nominal developmental stage, the morphology among individuals rarely looks exactly the same due to different imaging conditions and embryo rotations as well as idiosyncratic features resulting from external and internal noise[15–17]. In addition, numerous factors can influence the rate of embryogenesis, thus separating developmental stage from absolute developmental

[1]Systems Biology of Development, University of Konstanz, Konstanz, Germany. [2]Friedrich Miescher Laboratory of the Max Planck Society, Tübingen, Germany. [3]University Hospital and Faculty of Medicine, University of Tübingen, Tübingen, Germany. [4]Centre for the Advanced Study of Collective Behaviour, Konstanz, Germany. [5]These authors contributed equally: Nikan Toulany, Hernán Morales-Navarrete. [6]These authors jointly supervised this work: Murat Ünalan, Patrick Müller. ✉e-mail: murat.uenalan@uni-konstanz.de; patrick.mueller@uni-konstanz.de

time[18-26]. As a consequence of structural and temporal variation, characterization of embryonic development and the transitions between morphological states remains subjective. Computer-driven methods have been proposed to tackle this problem and to enable standardization by addressing structural or temporal variability[27-35]. However, approaches based on supervised machine-learning techniques require large databases, training resources and human-assisted annotation. Moreover, they admit only a limited number of predefined classes and therefore do not provide a generalizable method to characterize the multitude of rapid time-dependent developmental features in different phyla.

To address these challenges, we present a new approach to analyze developmental time by calculating the similarity between embryos of different timepoints. Our approach is based on Twin Networks, which can be used for the calculation of similarities between complex input vectors[36] with main previous applications in security verification tasks[37,38] and object tracking[39-41]. Using a high-throughput imaging pipeline, we first created a dataset comprising more than three million images with more than 15,000 zebrafish embryos. We then trained a Twin Network based on image triplets of normally developing embryos and applied the resulting model to accurately determine the developmental age of zebrafish. We applied our developmental age estimation approach to study how developmental tempo in zebrafish and medaka is affected by temperature, and found that classical physical biology theories[42,43] captured temperature-dependent development within a species-specific thermally adapted range. Moreover, we found that the Twin Network model can be used to characterize natural variability of zebrafish development and to robustly identify a small fraction of embryos that developed abnormally. Similarly, the Twin Network was able to detect small-molecule-induced phenotypic changes in embryonic development. Finally, we demonstrate that the Twin Network can be used to highlight key points of development, to describe transitions between stages and to automatically detect the main epochs of embryogenesis from developmental trajectories in an automated manner. Our method thus offers multimodal possibilities to analyze developing embryos with minimal previous knowledge about the process of interest and might also have widespread applications in other fields where complex processes unfold over time.

## Results

### Using similarity profiles to automatically stage embryos

Twin Networks consist of two identical parallel neural networks that share both architecture and weights to learn hidden representations of input data (Fig. 1a). These networks serve as the core for nonlinear dimensionality reduction of complex two-dimensional input matrices—such as images—to feature embeddings consisting of a series of numbers. Twin Networks compare images through similarity calculations based on feature embeddings, in contrast to classification algorithms that assign classes as two images are compared. We hypothesized that

the calculation of similarities between embryo images would allow to accurately account for complex morphological changes in silico in an unbiased manner. Therefore, we considered this model architecture to be ideally suited for the analysis of development.

We first used high-content microscopy to generate a dataset of more than 15,000 zebrafish embryos with high temporal resolution, covering the first day of development from cleavage to early larval stages (Extended Data Fig. 1a). A total of two million images was acquired, where each image position comprised up to 30 zebrafish embryos. We trained a ResNet101 deep learning model for image segmentation and zebrafish embryo detection with a positive predictive value of 99%. Application of this model to our experimental dataset combined with manual quality control facilitated segmentation into more than three million embryo image segments sorted by embryo and acquisition timepoint (Extended Data Fig. 1a). We then developed a Twin Network architecture designed to learn phenotypic features from triplets of image segments by training with triplet loss[44] (Extended Data Fig. 1b,c). This allowed us to calculate similarities between pairs of images by creating image embeddings and calculating the cosine similarity between them (Fig. 1a). By comparing two images of zebrafish embryos with the Twin Network, we obtained a similarity score for the compared individuals (Fig. 1a).

We reasoned that, if a test image of an embryo was compared with a set of other embryo images, the test image could be classified into similar embryonic phenotypes based on the similarity scores (Fig. 1b). We therefore used a timeseries of developing embryos as a reference with which a single test image was compared (Fig. 1c). The resulting graphs of similarities over time have two main characteristics relevant for our analysis. First, the peak of the curve, that is, the maximum similarity of the test embryo to reference images, reveals in which developmental stage the test image embryo is located (Fig. 1b). Repeated calculations of predicted developmental stages for a set of timeseries images of one embryo allow a trajectory based on predicted developmental stages to be constructed (Fig. 1d,e). Second, the nonpeak region of the curve contains additional information, such as the width of the peak (green box; Fig. 1b) and similarities to distant embryonic stages. These features are distinct at different timepoints and may resemble morphological similarity between unrelated developmental phases (for example, similarity of cleavage and blastula stages). Importantly, when comparing similarity curves of two images of an embryo taken a few minutes apart, the Twin Network attributes the successively acquired image to later stages by showing increased similarity values of the nonpeak part of the similarity curve to later developmental stages. Likewise, the difference between the similarity plots of these images is positive following the peak of the curve, indicating higher similarity to later developmental stages of the image that was acquired later (Extended Data Fig. 2). Furthermore, our Twin Network showed good precision in image ordering without a priori knowledge (Supplementary Note 2 and Supplementary

**Fig. 1 | Characterization of zebrafish development with Twin Networks. a**, Architecture of the Twin Network pipeline. **b**, Schematic for embryonic age prediction. A test image (top) is compared with a sequence of reference images with known temporal order (middle). The age of the test image corresponds to the age of embryo images with highest similarity (red dashed line); expectation based on absolute (gray curve) or relative (blue curve) similarity of input data. The width of the peak is indicated by green shading. **c**, Similarity plots between test embryos (top) and reference images (bottom). Each test embryo was compared with three reference image sets. The mean of cosine similarities to these reference sets is plotted as a datapoint for each reference image timepoint. Boxplots are based on the distribution of similarity values above 0.8. The center represents the median, box limits represent upper and lower quartiles, whiskers the 1.5× interquartile range and red points the curve maxima. Three images from the acquisition of one embryo, representative for three independent

experiments, are shown. **d**, Schematic for prediction of embryonic trajectories. Calculation of similarities between a test embryo (top) and a reference image timeseries (bottom row) with the peak of the similarity curve corresponding to the predicted embryonic age of the test image. Bottom row, left plot shows similarity curves calculated for several test images with known temporal order of one test embryo. For each image in the test image sequence, predicted embryonic age can be calculated. The right plot shows predicted embryonic age for each image in the test image sequence plotted based on the known temporal order of the test images. **e**, Developmental trajectory reconstructed for one representative test embryo ($n = 126$, see Extended Data Fig. 3e for median of full dataset). The blue scattered datapoints show the Twin Network prediction and the red line is the expected groundtruth trajectory. Test images are shown next to the $x$ axis, reference images are shown next to the $y$ axis. Scale bars, 500 μm.

Figs. 2 and 3). These analyses show that the Twin Network can be used to extract complex phenotypic fingerprints of embryos, which enables accurate automatic staging (Fig. 1e).

## Developmental tempo as a function of temperature

Temperature is a ubiquitous environmental factor that has a direct influence on developmental rates, affecting various aspects of an organism's

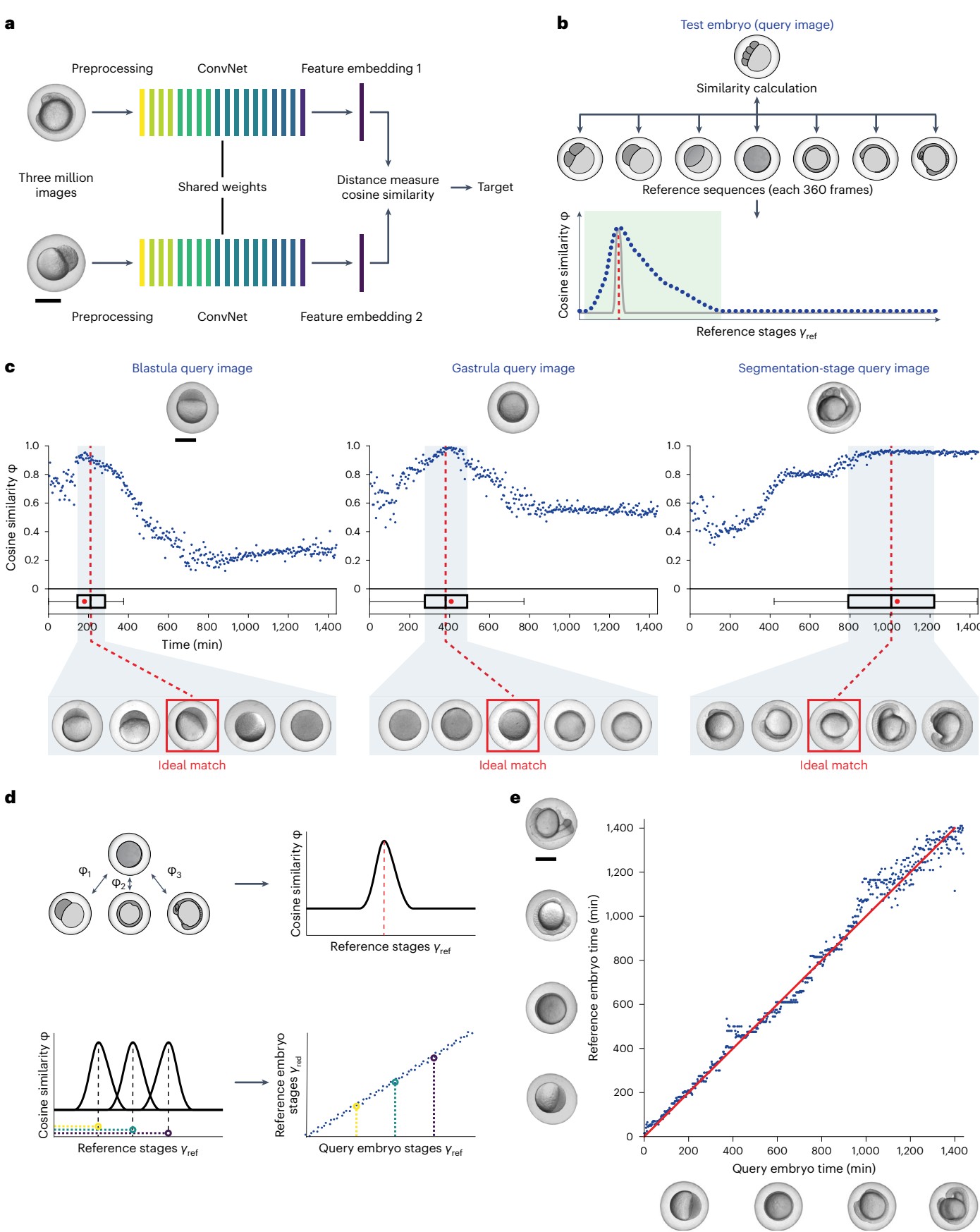

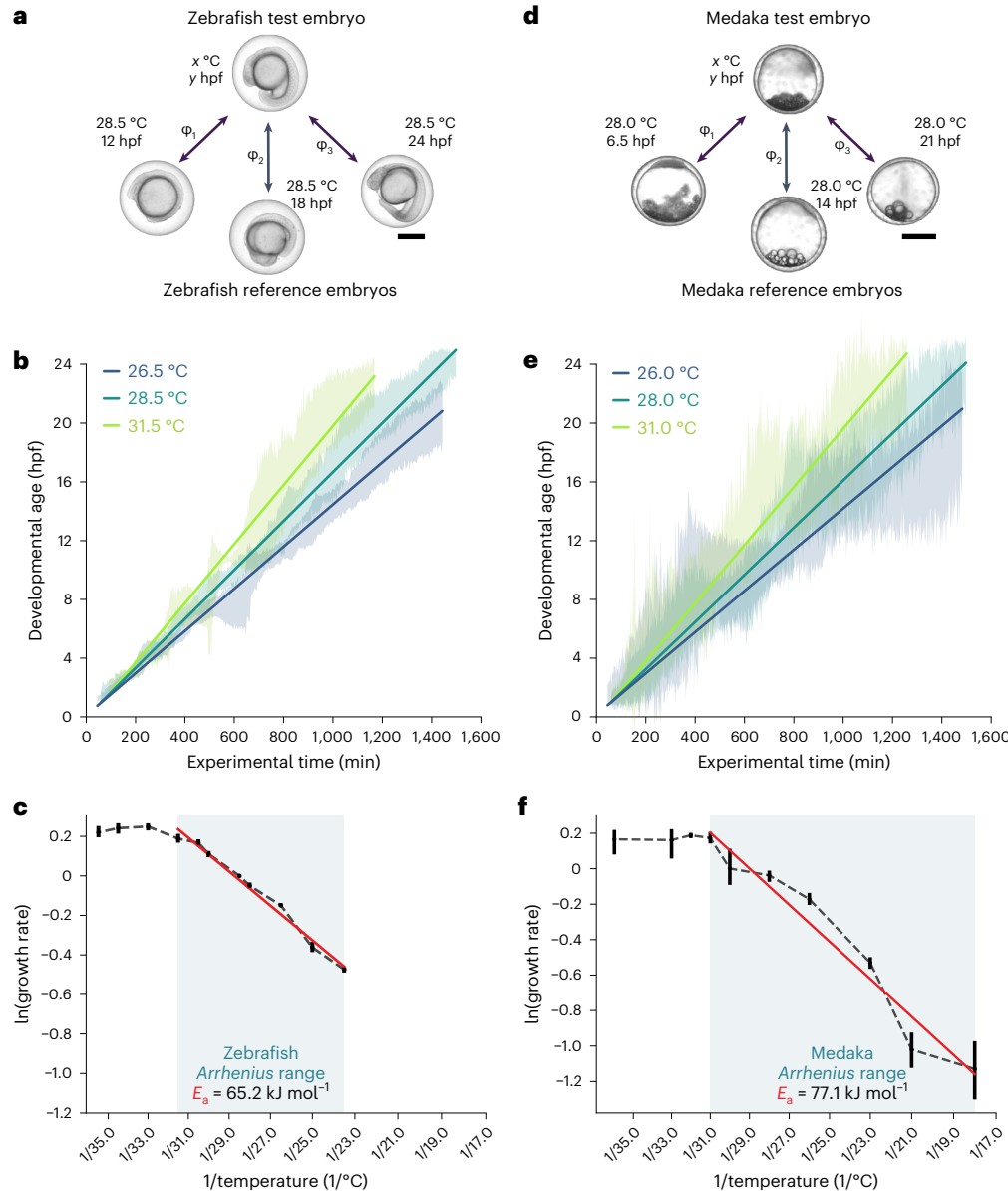

**Fig. 2 | Automated analysis of fish developmental temperature dependence using Twin Networks. a–f**, Analysis of zebrafish and medaka embryo development at various temperatures. **a**,**d**, Schematic for age estimation of zebrafish (**a**) and medaka (**d**): image of an embryo at a given timepoint (*y* hpf), raised at the temperature of interest (*x* °C), is compared with all timepoints (three examples are shown) at the reference temperature. Developmental age is assigned by the highest cosine similarity (*φ*). Scale bars, 500 μm. **b**,**e**, Developmental age estimation for zebrafish (**b**) embryos at 26.5 °C, 28.5 °C and 31.5 °C (*n* = 209, 126 and 130, respectively) and medaka (**e**) embryos at

26.0 °C, 28.0 °C and 31.0 °C (*n* = 47, 46 and 21), respectively. Error envelopes represent two times the median absolute deviation (MAD) over the embryos and are shown together with the corresponding linear fit (solid line). **c**,**f**, Natural logarithm of the estimated growth rates for zebrafish (**c**) and medaka (**f**) at various temperatures. Error bars represent 99.99% confidence intervals from bootstrapping with 100 repetitions around the estimated slope of the linear fit to the data shown in Extended Data Figs. 3 and 4. Blue shading shows the *Arrhenius* range; the apparent activation energy is stated.

life cycle from reproduction to ecological distribution[45–47]. Understanding the temperature dependence of embryogenesis can provide valuable data for developmental biology, offering new insights into the underlying molecular and physiological mechanisms that orchestrate the early stages of life[21,48–52]. This not only sheds light on the adaptive strategies employed by different species in diverse environments but also provides critical knowledge for predicting the impacts of climate change on natural populations and ecosystems[45,53].

Previous efforts to quantify the temperature dependence of embryonic development involved manual or semiautomated annotation of developmental time, limiting the number of experiments

that could be analyzed in a reasonable timespan[51,54,55]. Recent work has shown that machine learning can be used to automate this process and distinguish zebrafish embryos developing at 25.0 °C and at 28.5 °C (ref. 33). To test whether our Twin Network could be used for automated analysis of temperature-dependent shifts in developmental tempo, we analyzed zebrafish embryos between 23.5 °C and 35.5 °C as well as evolutionarily distant medaka embryos that can tolerate a wider temperature range from 18 °C to 36 °C. The lower end of the temperature range was chosen because medaka embryos arrest below 15 °C (ref. 56), and zebrafish did not survive below 23 °C (Supplementary Video 2)[54,57]. For each temperature condition, we analyzed between 100

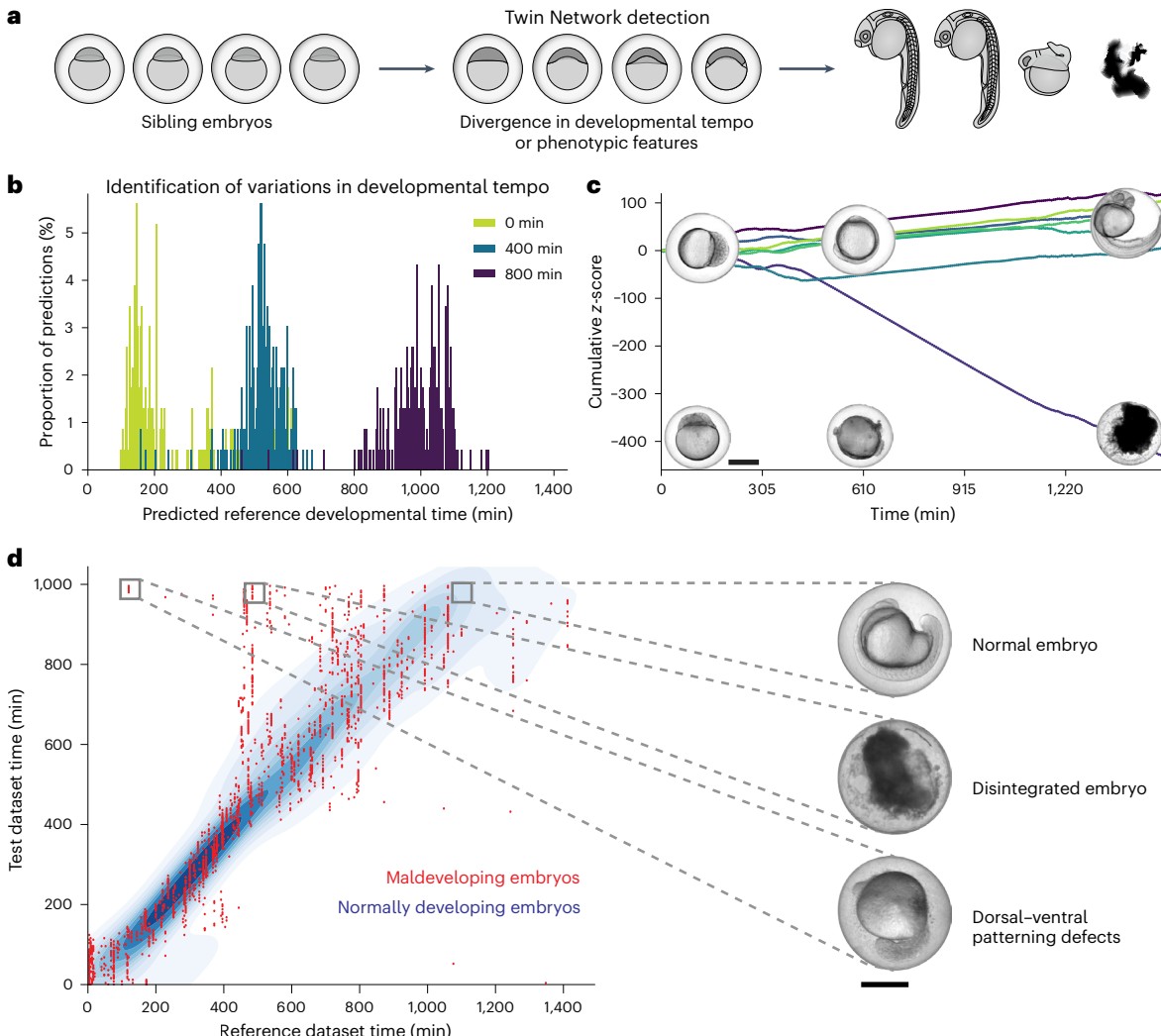

**Fig. 3 | Detecting morphological variability during zebrafish embryogenesis and deviation from normal development. a**, Siblings at the same nominal age may display a wide variety of morphologies due to differences in developmental tempo or differences in expression of features. **b**, Percentages of predicted developmental stages when comparing 77 embryos at 0 min (green), 400 min (blue) and 800 min (purple) after acquisition start with a reference embryo set with images of embryos at the age of 0.2–24.2 hpf. **c**, A decrease in average similarity values of a single embryo indicates phenotypic deviation, for example, due to spontaneous maldevelopment. The sum of average similarity $z$-scores for seven embryos is shown, each represented by a colored line. Images of a normal (top) and a defectively (bottom) developing embryo are shown at corresponding acquisition timepoints. **d**, Predicted stages of abnormally developing embryos (red) frequently fall outside of the normal range of predicted stages for 14 reference embryos (underlying density plot). $n$ (maldeveloping embryos) = 7, $n$ (normally developing embryos) = 14. On the right, a normal embryo at 17 hpf, a dissociating embryo and an embryo with dorsal–ventral patterning defects, each at 24.2 hpf, are shown with reference to the endpoints of the corresponding analyzed developmental trajectories. Scale bars, 500 μm.

and 200 zebrafish embryos or between 20 and 100 medaka embryos, ensuring robustness and reliability in our analysis (Fig. 2). We utilized a Twin Network trained exclusively on embryos at a reference temperature for each species (28.5 °C for zebrafish and 28.0 °C for medaka).

Classical physical biology theories predict that reaction rates scale with temperature[42,43]. Indeed, developmental tempo varied profoundly at different incubation temperatures for zebrafish and medaka embryos: Whereas at lower temperatures embryonic development proceeded at a slower pace, higher temperatures elicited a marked acceleration in development compared with the reference temperature (Fig. 2a,b,d,e, Extended Data Figs. 3 and 4 and Supplementary Video 3). Strikingly, zebrafish and medaka adjusted their developmental tempo by a factor of approximately two when subjected to a temperature change of 10 °C—in good agreement with the $Q_{10}$ rule of thumb for chemical reactions[58]. To analyze temperature-dependent developmental tempo more quantitatively, we used the Twin Network

to estimate the growth rate for different temperatures and fitted the data with the classical Arrhenius equation[43]. From the slope of the linear fit within a species-specific range of temperatures, we estimated apparent activation energies of 65 kJ mol[−1] for zebrafish and 77 kJ mol[−1] for medaka, comparable with other poikilotherm organisms like frogs, flies or yeast—and notably different from homeotherms like mice or humans[55,59] (Fig. 2c,f). Interestingly, the temperature ranges correlated with the temperatures that support normal development in these fish species, in accordance with the notion of the *Arrhenius* range that refers to the spectrum of temperatures in which regular growth and biochemical reactions of specific organisms scale with temperature[59]. However, at higher temperature regimes the developmental rate no longer accelerated but instead stabilized, displaying intriguing deviations from the idealized theories (Fig. 2c,f). A similar behavior has been found in *Drosophila* and might reflect a reaction to heat stress[51]. Interestingly, the two species that we analyzed reacted differently to

temperatures at the lower edge of their comfort zone. Zebrafish development slowed down linearly (Extended Data Fig. 3), and temperatures below 23 °C were lethal. Medaka embryos, on the contrary, displayed a nonlinear development—that is, initial linear development followed by a partial arrest—at the two coldest temperatures analyzed, spending a disproportionately long timespan in blastula stages (Extended Data Fig. 4a,b). These findings underscore the importance of automated techniques in comprehending intricate biological phenomena, opening new possibilities for further research and application in diverse biological systems.

## Quantifying natural variability during embryogenesis

Animal development is a remarkably reliable process that consistently results in a complete embryo despite genetic variation, external perturbations and the noise and stochasticity associated with gene expression[60–63]. However, even if embryos were laid at the same time and incubated under the same conditions, growth rates may vary between embryos and can lead to deviations in developmental stages over time[6] (Fig. 3a and Supplementary Videos 4–6).

To test whether this divergence of individual phenotypes in an ensemble of similarly aged sibling embryos can be detected by our Twin Network, we calculated similarities to reference images for several embryos of similar age. We found that, for several siblings laid at the same time, the early stages of embryonic development predicted with our Twin Network had a narrow distribution (green; Fig. 3b and Extended Data Fig. 5a,b). Interestingly, and consistent with expert human assessment[6], the distribution width of predicted embryonic stages increased after the beginning of the segmentation period (blue and purple; Fig. 3b and Extended Data Fig. 5a,b), whereas average similarities decreased during embryonic development (Extended Data Fig. 5a,b). These results show that our Twin Network can be used to quantify even small and fine-grained developmental changes as well as natural variability during embryogenesis.

In contrast to these small variations, developmental robustness can fail in a fraction of abnormally developing embryos[64,65]. Indeed, in our dataset of more than three million zebrafish embryo images, we found that 1% of the embryos developed abnormally, frequently due to spontaneous disintegration or dorsal–ventral patterning defects[66,67] (Supplementary Videos 4–6). To test whether such naturally occurring phenotypes can also be detected by our Twin Network, we first used trajectories of aphenotypic embryos to define a normal range of predicted developmental stages for each acquisition timepoint (Fig. 3c,d). Strikingly, embryos identified to be abnormal by a human scientist frequently deviated from this normal range much earlier (Fig. 3c,d). Based on low average similarity values, abnormally developing embryos could be detected in a batch of sibling embryos at early stages (Fig. 3c). It will be interesting in the future to use this approach combined with genomics, transcriptomics and proteomics techniques as a tool to reveal the molecular details of why robust development fails in these deviating embryos.

## Identifying drug-induced embryonic phenotypes

Embryonic development is coordinated by signaling molecules, and modulating their activity can cause characteristic phenotypic changes[68]. During zebrafish development, seven main signaling pathways play a pivotal role in coordinating the establishment of the body plan. While germlayer patterning and the formation of anterior–posterior and dorsal–ventral axes are regulated largely by bone morphogenetic protein (BMP), retinoic acid (RA), Wnt, fibroblast growth factor (FGF) and Nodal signaling, the elongation and morphogenesis of the body axis is under strong control of the sonic hedgehog (Shh) and planar cell polarity (PCP) signaling pathways[69]. When the activity of any of these pathways is modulated, distinct patterning defects emerge. We recently developed a deep learning-based classification algorithm—EmbryoNet—trained with manually annotated images to

detect such defects and link them to one of the main embryonic signaling pathways[31]. This classification approach used a finite number of predetermined classes. We reasoned that Twin Networks could be used to detect abnormally developing embryos without predefined classes, and instead detect deviating embryos based solely on similarity scores. This would enable unbiased automated analyses of large-scale drug screens to discover compounds that potentially elicit new phenotypes or intermediate phenotypes between previously defined classes.

To test the utility of Twin Networks in the detection of abnormal embryos, we compared the phenotypes of untreated embryos with those of embryos treated with BMP, Nodal, FGF, Shh, PCP and Wnt inhibitors as well as RA exposure (Fig. 4). We used the Twin Network to compare groups of embryos of each condition with a reference group of untreated embryos over time (Fig. 4a). Comparison of embryos in the untreated group revealed high similarity values (Fig. 4b), indicating coherence within a developmental cohort. In contrast, similarity values between untreated and small-molecule drug-treated embryos were consistently lower for most of the treatments (Fig. 4c–i and Supplementary Videos 7–13). Next, we analyzed the differences statistically to identify the timepoints at which the group of embryos deviated significantly from the reference. This allowed us to detect groups of embryos with phenotypic defects without previous knowledge of the specific alteration. The accuracy of detection depended on the number of analyzed embryos and the type of perturbation (Fig. 4j).

To determine how accurately our method can identify phenotypes with different levels of penetrance and severity, we used the well-characterized phenotypic spectrum in zebrafish embryos with different levels of BMP pathway inhibition, resulting in the previously defined classes C2, C3, C4 and C5 with increasing degree of dorsalization[70]. *bmp* mutants and highly penetrant phenotypes resulting from treatment with high doses of small-molecule BMP signaling inhibitors required only a few embryos for accurate detection of developmental deviations, and milder phenotypes could be detected with a larger number of ~30 embryos (Extended Data Fig. 6 and Supplementary Videos 14–18). These analyses show that the Twin Network—which had previously been trained only with images of normally developing embryos—can detect phenotypic changes in an unbiased manner.

## Automated derivation of developmental epochs

Images of reference embryos can be used to assess the developmental timing of a test embryo (Fig. 1b–e), but such reference images are not always available, for example, for newly discovered or uncharacterized species. Another way to characterize a developmental process with minimal previous knowledge is to calculate the similarities of a test image to other images of the same embryo at earlier timepoints (Fig. 5a).

To test this idea, we calculated similarity profiles in this manner for zebrafish embryos, which resulted in distinct similarity profiles at different development times (Fig. 5b). We noted a common pattern, where high similarity values were clustered locally; in contrast, similarity values at more distant timepoints were lower and formed plateaus (Fig. 5b). Interestingly, the local and global statistical similarity of image pairs measured by the network were coherent with the sequence of key stages during development; embryos at timepoints that fell into an extended plateau were characterized by stable morphologies (Fig. 5b), highlighting principal developmental epochs such as the classical cleavage, blastula, gastrula, organogenesis and segmentation stages[6]. In contrast, embryos at timepoints that fell into a boundary between plateaus represented short-lived epochs with principal changes in developmental morphologies (Extended Data Fig. 7 and Supplementary Fig. 4). Thus, the Twin Network allows the automatic generation of staging atlases akin to human assessment, but de novo, without previous knowledge of the developmental stages and without a model that was specifically trained for this purpose.

We next asked whether this approach to generate species-specific staging atlases in an automated manner could be generalized. We first

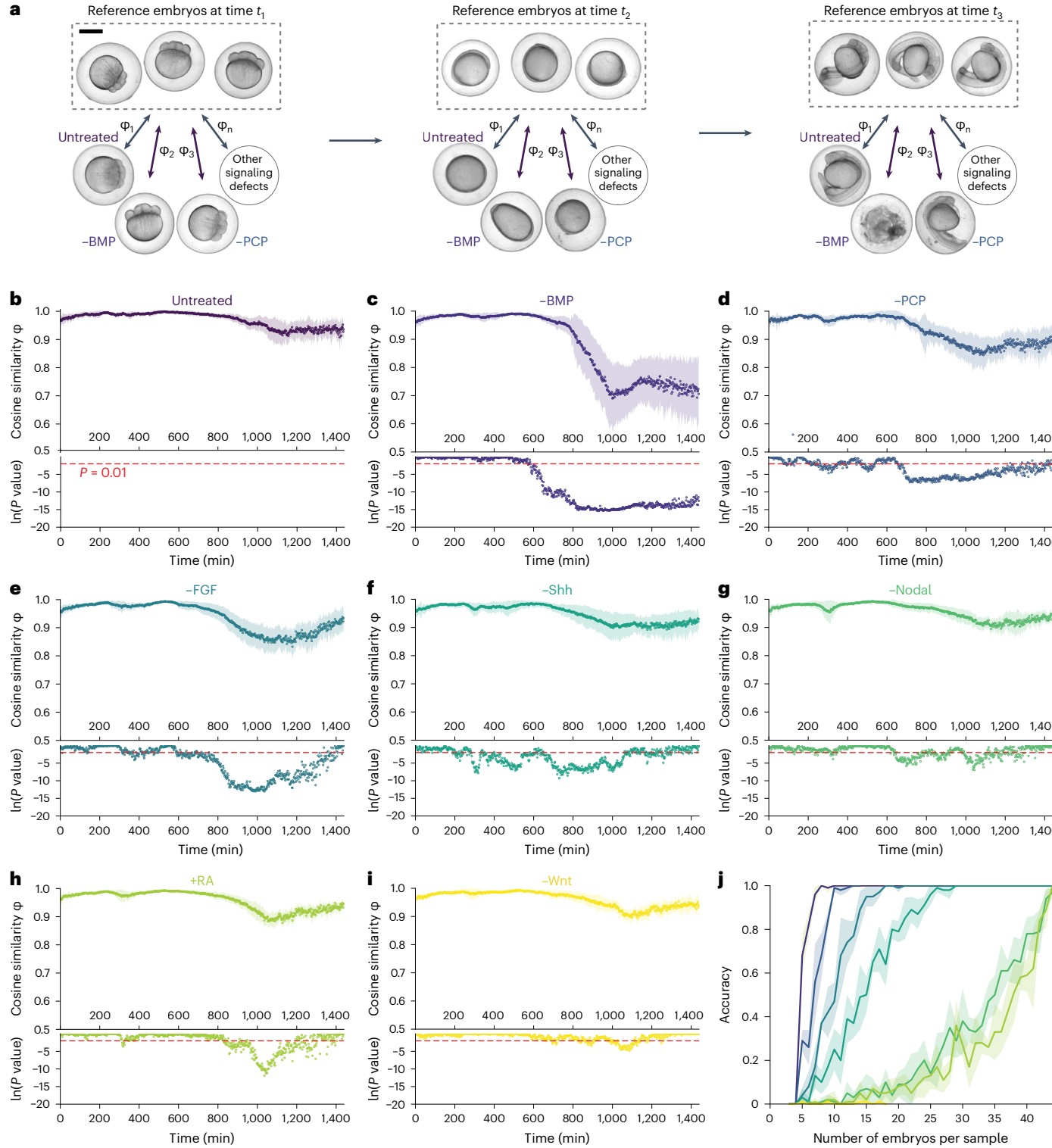

**Fig. 4 | Application of Twin Networks to identify drug-induced phenotypes.**
**a**, Strategy of similarity calculation between embryos at the same developmental stage under different drug treatments. An untreated embryo (top) serves as reference to which drug-treated embryos (bottom) are compared. Examples for untreated, BMP-inhibited and PCP-inhibited embryos are shown at 1.25, 10 and 26 hpf. The cosine similarity between a treated embryo and the reference embryo is calculated for every timepoint. Scale bar, 500 μm. **b**–**i**, Upper panel, mean similarities and s.d. of similarities for untreated ($n = 44$) (**b**) and −BMP ($n = 44$) (**c**), −PCP ($n = 14$) (**d**), −FGF ($n = 44$) (**e**), −Shh ($n = 44$) (**f**), −Nodal ($n = 44$) (**g**), +RA ($n = 44$) (**h**) and −Wnt ($n = 18$) (**i**) embryos relative to the reference group of untreated embryos as a function of time. Lower panel, significance levels of the difference from untreated embryos determined using a nonparametric one-sided Mann–Whitney $U$ test over each timepoint of the image series. No adjustments for multiple comparisons were made. **j**, Dependency of the accuracy of abnormality detection on the number of embryos used to analyze −BMP, −PCP, −FGF, −Shh, −Nodal, +RA and −Wnt embryos. Mean and s.d. are shown for five repetitions with randomly selected samples.

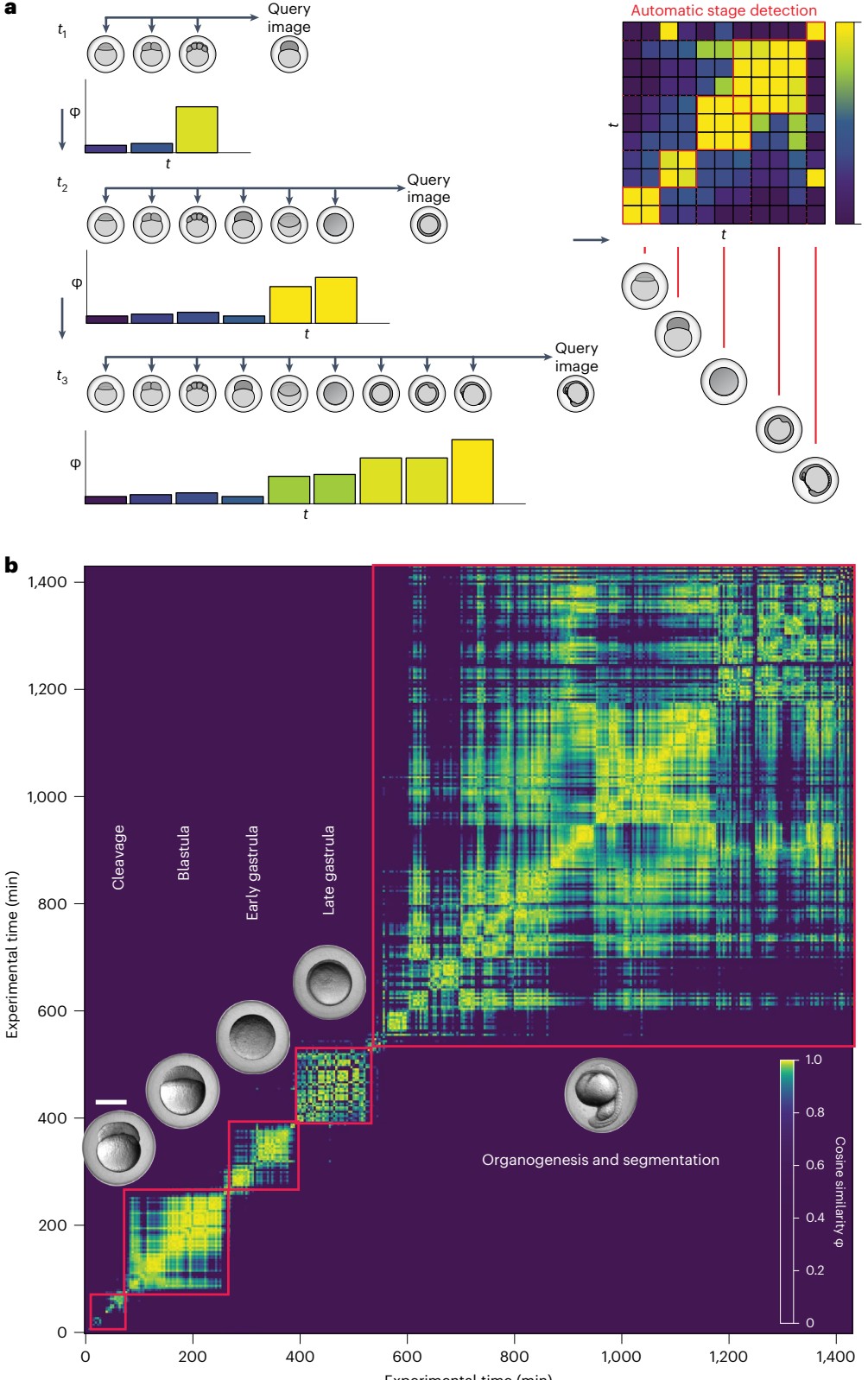

**Fig. 5 | Automatic detection of developmental epochs. a**, Strategy for the calculation of similarities between a test embryo and images from previous acquisition timepoints of the same embryo. Left, for each embryo image, the cosine similarity with all the images from previous acquisition timepoints is calculated. Given the symmetry of pairwise comparison, such similarity profiles can be stored in a symmetric squared matrix (right) that describes the self-similarity of an embryo during development over time. **b**, Representative similarity matrix showing pairwise comparison of an image series of one zebrafish embryo with itself. High similarity between neighboring timepoints defines clusters that correspond to developmental epochs. The Twin Network detects and partitions embryo development into phases that are in line with the classical zebrafish staging atlas[6]. Imaging was started at 2 hpf (64-cell stage). Full analysis shown in Extended Data Fig. 7a–c. Scale bar, 500 μm.

addressed this question with two other fish species—medaka (*Oryzias latipes*) and three-spined stickleback (*Gasterosteus aculeatus*)—that had diverged from zebrafish (*Danio rerio*) hundreds of millions of years ago[31]. When applied to timeseries of these morphologically diverse embryos, the Twin Network yielded an informative atlas for each embryo (Extended Data Figs. 8 and 9). We then extended this approach to an even more distant taxon represented by the nematode *Caenorhabditis elegans*. We used open data available from different independent sources such as published papers[71] and YouTube videos for training and evaluation, respectively. This allowed us to identify the first cleavage cycles automatically, giving rise to the first four blastomeres in *C. elegans* (Extended Data Fig. 10).

These results show that the Twin Network approach can be used to determine staging atlases de novo for different organisms and using a broad range of size and quality of image datasets.

## Discussion

Here we present a machine-learning-based approach to describe developing processes in an automated and objective manner. The central element of our approach is the unsupervised computation of similarities between states. Our model can be applied to multimodal tasks in the analysis of animal development and compares favorably with classical vector diffusion maps for image registration in terms of precision.

Our Twin Network results have four main implications. First, our approach provides a standardized way to stage and compare embryos. Accurate estimation of an individual's age is important for any developmental biology study because research results may vary at different embryo stages. However, phenotypic transitions can be very fluid, and it is often difficult to relate an observed embryo to the idealized description in staging atlases. Our Twin Network approach takes into account the smooth transitions between developmental stages, where phenotypic traits may appear at one point in development and persist or disappear at another timepoint. By performing systematic similarity calculations of a test image with a reference image sequence, we retrieve a similarity plot that can be used to accurately assign an embryo to a range of developmental steps within the reference sequence. Depending on the length of the reference sequence, this can be done within seconds on a GPU-based workstation. It seems that our Twin Network learns to dynamically represent phenotypic traits and combine them for similarity computations at different developmental stages, instead of creating static sets of features for distinct classes of phenotypes. Furthermore, our Twin Network is able to point a theoretical arrow-of-time that represents the developmental direction.

Second, we found a tight connection between ambient temperature and developmental tempo in agreement with predictions from classical physical biology theories[42,43]. Apparent activation energies of zebrafish and medaka are on the order of ~60–70 kJ mol$^{-1}$, potentially making their enzymatic reactions highly efficient even at lower temperatures[59]. It is tempting to speculate that this range of metabolic rates is optimal to adapt to a diverse array of temperatures. In contrast, mammalian cells—being more specialized and sensitive to environmental changes—have evolved with narrower *Arrhenius* ranges. This trait enables them to function optimally within specific temperature limits, but it also comes at the cost of higher apparent activation energies of 120 kJ mol$^{-1}$. This higher energy requirement could be important for maintaining the intricacies of cellular processes at warmer temperatures[59]. Our findings provide support for the notion of an inverse relationship between *Arrhenius* ranges and apparent activation energies across different taxa. Interestingly, in contrast to zebrafish embryos with a sharp lower temperature limit, medaka embryos nonuniformly slowed down at colder temperatures. It is conceivable that this nonuniformity is the basis for the medaka embryos' ability to arrest development below 15 °C for up to 3 months[56,72]. These findings shed light on the evolutionary strategies

adopted by various organisms to cope with temperature fluctuations and highlights the interplay between temperature adaptation and biochemical kinetics[59].

Third, our approach enables the detection of phenotypic variability within a population. We parametrized the divergence of features using similarity scores as indicators of temporal and feature deviations. Using our Twin Network, we found that variability increased over the course of embryonic development. Even though our Twin Network was trained only on images of normally developing embryos, it also detected spontaneous as well as small-molecule-induced malformations. This shows that the Twin Network is impartial to the specific treatment and robustly identifies embryos that deviate from normal developmental trajectories. The Twin Network approach might therefore be ideally suited to study embryonic phenotypes associated not only with one, but also with combined signaling defects, extending our previous approach to investigate embryonic phenotypes associated with signaling defects[31].

Fourth, Twin Networks can be used to automatically generate atlases of the main epochs during development in diverse species. Large areas of similarity correspond phenotypically to principal developmental phases, and smaller areas correspond to a finer subdivision of embryogenesis into developmental steps. Thus, development is characterized by the stereotypic alternation of periods, in which embryonic morphologies change, and phases, in which embryonic morphologies undergo little change. Strikingly, this allows essential developmental epochs in the course of embryogenesis to be identified on the basis of a single individual in an unsupervised manner for different specimens, amount of training data and quality of images. We expect that this approach will be widely applicable and useful to describe the development of uncharacterized species and to facilitate their use in studies of development and evolution. A current limitation is that a direct application of our models to different image data (for example, different species, different imaging conditions) is not possible. However, this could be achieved by fine-tuning or retraining the models to adapt them to specific applications. Moreover, more general and robust models could potentially be generated by future methodological improvements such as taking advantage of Generative Adversarial Networks to create expansive datasets when experimental data is scarce.

In summary, Twin Networks can capture complex systems and map several facets of their development by computing similarities between images. Developmental time can be accurately measured de novo, allowing unbiased quantitative studies of robustness from limited visual cues. In general, precise and objective assessment of phenotypic traits in spite of several sources of variation is not only necessary for the description of embryogenesis, but a principal problem in many fields of biology and beyond where Twin Network applications can provide new insights.

## Online content

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

## Methods

### Sample preparation

Zebrafish ages ranged from 2 months to 2.5 years at the time of mating. Embryos at the one- to eight-cell stage were obtained from matings between two to five female and male zebrafish. Fertilized embryos were selected manually using a glass Pasteur pipette. Selected embryos were washed three to five times with 200 ml embryo medium and kept in the same medium before microscopy. Embryos were transferred to 1-, 6-, 24- or 96-well plates (Greiner Bio-One) for microscopy in embryo medium or 1% low melting-point agarose in embryo medium[31,73]. Depending on the size of the plates, 5–100 embryos on average were placed in multititer plates. During microscopy of embryos, plates were covered with transparent Saran wrap to prevent medium evaporation. To maximize the utility of our approach for different genetic backgrounds, we used a variety of aphenotypic zebrafish lines: TE (ref. [74]), *Tg(sebox:EGFP)*[75], *Tg(gsc:GFP)*[76], *Tg(gsc:TurboRFP)*[77], *Tg(lhx1a:EGFP)*[78] and *sqt*[+/−] (ref. [79]). An overview of zebrafish crosses used to acquire embryo timeseries is given in Supplementary Table 1. For the temperature experiments, zebrafish eggs were collected within 15 min after mating and distributed into multiwell plates with embryo medium. Medaka eggs of the Cab strain were collected from standard crosses into cold medaka embryo medium (17 mM NaCl, 0.4 mM KCl, 0.27 mM $CaCl_2$, 0.65 mM $MgSO_4$) to synchronize them at stage 1 (ref. [7]). Adhesive filaments were removed with sandpaper, and the separated embryos were distributed into multiwell plates with temperature-adjusted medaka embryo medium.

### Image acquisition

Images of zebrafish embryos for training and aberrant phenotype analysis were acquired using an ACQUIFER Imaging Machine (ACQUIFER Imaging GmbH) with a 12-bit Hamamatsu sCMOS 2k × 2k sensor (Hamamatsu Photonics) and a ×2 magnification objective (Nikon) controlled by Imaging Machine control software (Acquifer Imaging GmbH, v.ID 4.00.21). Imaging was performed with the acquisition parameters listed in Supplementary Table 1 at intervals of 2.0–8.3 min at 28 °C. Each well was recorded as a separate image stack for 0.25–25 h, resulting in 3–720 acquisition timepoints depending on the acquisition interval. In total, more than 2 million images were acquired and quality-controlled in 52 separate experiment runs, from which 34 experiments were selected manually for image quality. These images were stored as 12-bit TIFF-files with 2,048 × 2,048 pixels (0.31 pixels µm$^{-1}$) in separate files with each image displaying 1 to 30 embryos.

The temperature series were acquired on two Keyence BZ-X810 microscopes with ×2 apochromate objectives, 3.7 W LED light sources and the BZ-X800 viewer software (Keyence, v.01.03.00.01). The embryos were imaged in 48-well plates (Eppendorf, catalog no. 0030723112). The microscopes were set up in a temperature-regulated room. Empty wells and the space between wells were filled with filtered water to help buffer the temperature. For one system, the experimental temperature was determined by the room temperature as measured by a ShT4x SmartGadget (Sensirion) directly next to the multiwell plate and a custom-built dipping thermometer in a reference well within the plate. Experiments outside ±0.5 °C of the target temperature were excluded. To image two temperatures in parallel, the second system was equipped with a heated chamber (H301-KEYENCE-BZX) with an UNO Stage top incubator thermal regulator (Okolab) and a multiwell frame providing a thermal uniformity of 0.3–0.4 °C. Zebrafish embryos were imaged every 2–5 min for 24 h, and timeseries from temperatures above 28.5 °C were truncated after the prim-6 stage[6]. Medaka embryos were imaged every 2 min for 24 h, and timeseries from temperatures above 28 °C were truncated after stage 19 (ref. [7]). Varying starting points of the timelapse videos were corrected by the experimental ages of the first timepoint. The exposure time was 0.13 ms with 50% relative intensity and 60% aperture stop. Images were stored as 8-bit JPEG files with 1,920 × 1,440 pixels (0.33 pixels µm$^{-1}$).

For drug-treated zebrafish embryos as well as medaka and three-spined stickleback embryos, open-source image data was used (https://doi.org/10.48606/15)[31]. For *C. elegans*, tiff images for training and testing were extracted from published videos[71] and https://www.youtube.com/watch?v=M2ApXHhYbaw, respectively. A total of 232, 56 and 1 embryos were used for training the models of medaka, stickleback and *C. elegans*, respectively.

### Image segmentation: preparation of the segmentation model

For detection and segmentation of zebrafish embryos in microscope images, an object detection model was trained using TensorFlow Object Detection API (TensorFlow v.2.2.0). An SSD ResNet101 v.1 FPN 640 × 640 (RetinaNet101) architecture, pretrained on the COCO dataset (https://github.com/tensorflow/models/blob/master/research/object_detection/g3doc/tf2_detection_zoo.md), was used as the object detection model. For training and testing, 877 images displaying embryos from blastula to 12-somite stages were selected manually. Embryo segments were annotated manually using Visual Object Tagging Tool (https://github.com/microsoft/VoTT, v.2.2.0). Training TensorFlow record files were created using a custom script (https://github.com/TannerGilbert/Tensorflow-Object-Detection-API-Train-Model/blob/master/generate_tfrecord.py). Training was performed according to the TensorFlow Object Detection API documentation (https://tensorflow-object-detection-api-tutorial.readthedocs.io/en/2.2.0/training.html). Evaluation of segmentation accuracy was performed manually using 36 test images containing 230 embryos. Segmented images and individual embryo tracking results were stored in separate JSON files for each analyzed image. Individual image segments were retrieved from the original acquisition images, and all embryo segment images were stored separately with information on acquisition timepoints for further usage.

For the analysis of developmental temperature dependence, single embryos were segmented using EmbryoNet (ref. [31]) and exported with a custom-built Matlab-script. After image acquisition and segmentation, the segmented timeseries of single embryos were loaded into Fiji (ImageJ v.1.54f)[80]. Unfertilized, dead or malformed embryos were excluded manually. For the identification of drug-induced embryonic phenotypes, image data and the corresponding segmentations were retrieved from (ref. [31]) and exported as single embryo images.

### Dataset cleaning

Acquired images were evaluated manually and put into different categories: normal embryos, embryo images that were out of focus or overlaid, disintegrating embryos and embryos displaying other abnormal phenotypes. Using a custom Python script, all embryos within these categories were divided into subgroups by checking for segment brightness, segment size and number of timepoints acquired for each single embryo. Dataset cleaning was performed to select high-quality images of embryos for model training. This classification resulted in a total of ten categories. The cleaning step resulted in a dataset of more than 3 million image segments, from originally 15 million acquired images. For each experiment, a separate JSON file was created containing information for embryos belonging to each category.

### Twin Network model training

The Twin Network architecture was based on the architecture of a vanilla Siamese Network (https://github.com/keras-team/keras-io/blob/master/examples/vision/siamese_network.py). A ResNet50 architecture with pretrained weights based on the ImageNet dataset (https://www.tensorflow.org/api_docs/python/tf/keras/applications/resnet50/ResNet50?hl=de) was used as backbone network for the embedding model of the Twin Network. The output of the ResNet50 backbone network was flattened and passed to a custom model head consisting of three dense layers with interposed batch normalization and an output/embedding size of (1, 256). For transfer learning, all

layers of the ResNet50 backbone network were frozen, except for layers of convolutional block 5 and the model head. ResNet50-generated feature embeddings were combined within a distance layer to calculate the Euclidean distance between network-generated embeddings of different inputs during the training process.

In each training step, three embryo images were combined into an image triplet and passed to the Twin Network: first, an image from a random developmental stage $t_1$ as 'anchor' image, second an image from a similar developmental step $t_1$ (model version 1) or the anchor image with applied image augmentation (model version 2) as 'positive' image and third an image from another developmental step $t_2 \neq t_1$ than the first image as 'negative' image. For zebrafish, two versions of the Twin Network model were trained, the first with 300,000 image triplets for ten epochs, and a second with 1,000,000 image triplets for two epochs. Triplet loss was applied to the model to minimize the distance between the embeddings of the anchor and positive image and to maximize the distance between the anchor and negative image. The loss for each image triplet passed to the network was calculated by

$$L(A, P, N) = \max(0, \|f(A) - f(P)\|^2 - \|f(A) - f(N)\|^2 + a)$$

with $A$ representing the anchor image, $P$ representing the positive image, $N$ representing the negative image, $f$ representing a function generating an image embedding and $a$ representing an additional margin for increased contrast between the distance of $A$ and $P$ and the distance of $A$ and $N$. The minimization of the resulting cost was performed by reducing the value of $\|f(A) - f(P)\|^2 + a$ and increasing the value of $\|f(A) - f(N)\|^2$.

Training was performed with GPU-acceleration using an NVIDIA GeForce RTX3070 (ASUS). Training duration was approximately 18, 12, 10 and 2 h for the models of zebrafish, medaka, stickleback and *C. elegans*, respectively.

The models for the analysis of developmental temperature dependence were trained with 1,000,000 and 100,000 image triplets for 40 and 70 epochs for zebrafish and medaka, respectively, using model version 1. Only data at the corresponding reference temperatures, that is, 28.5 °C and 28.0 °C for zebrafish and medaka, respectively, were used for the training. To evaluate the variability of the predictions for the similarity matrices, ten models were trained using 100,000 image triplets (from the training set of the temperature analysis of zebrafish) for 40 epochs. The models for medaka, stickleback and *C. elegans* were trained for 30 epochs using 150,000, 150,000 and 100,000 image triplets, respectively, using model version 1. These trainings were performed with GPU-acceleration using an NVIDIA GeForce RTX3090 graphics card (ASUS).

### Similarity calculation between images
For further similarity calculations, the trained ResNet50 model was used to generate embeddings of given images. Model-generated embeddings were used to calculate cosine similarities between two inputs, hereby returning a numeric estimation of the concordance between two images. Cosine similarity was calculated as follows:

$$\text{cosine similarity } \varphi = \frac{\mathbf{a} \cdot \mathbf{b}}{\|a\|\|b\|} = \frac{\sum_{i=1}^{n} a_i b_i}{\sqrt{\sum_{i=1}^{n} a_i^2}\sqrt{\sum_{i=1}^{n} b_i^2}}$$

### Image comparison types
Similarity calculations were performed based on different test and comparison images and image sequences. A complete overview of performed comparisons is shown in Supplementary Table 2. Two different types of reference images were used in the example applications of Twin Network to embryonic development: reference images were selected either as a distribution of different acquisition timepoints, representing different developmental stages, or at the same acquisition timepoint as a distribution of different phenotypic characteristics. Reference images from different acquisition timepoints were used to predict developmental stages, establish developmental trajectories, determine developmental epochs and detect abnormal development based on deviations in predicted developmental stages. Reference images from similar imaging timepoints were used to illustrate variability in embryonic phenotype, to predict the effects of chemical compounds on embryonic phenotype and to detect spontaneous maldevelopment during embryogenesis.

### Image sorting
A set of $n$ images to be ordered was passed to the trained ResNet50 architecture, and $n$ image embeddings were generated. Euclidian distances and cosine similarities between all $n$ embryo embeddings were calculated; $z$-scores were calculated for both distance metrics, and $z$-scores of Euclidian distances were subtracted from $z$-scores of cosine similarities. The embedding index with the overall highest similarity $z$-score to any other embedding was selected as start index. Beginning at the embedding value with the start index, for the next index the index with the highest $z$-score of the start index was selected. This process was iteratively repeated until all indices were assigned an order index. Each time an index was selected, the index was removed from a list of available indices. In case that the index with highest similarity to the last index was already assigned an order index, the index with second highest, third highest and so on, similarity value was selected.

For the comparison of the Twin Network and classical vector diffusion map-based image ordering, a Kolmogorov–Smirnov test was first performed to check whether the absolute deviations from the groundtruth were distributed normally for both approaches. A two-sided Wilcoxon signed-rank test was used to compare whether the difference in non-normally distributed data between the two methods was significant.

### Developmental stage and epoch prediction
For prediction of developmental stages of zebrafish embryos, similarities were calculated between images of a test embryo from 0.5–2.0 to 24–25 h postfertilization (hpf) and reference embryos at different developmental timepoints. One image of the test embryo was compared with an image timeseries with $n$ images of ten reference embryo anchors, where for each image the acquisition timepoint was known. The ten embryo anchors were selected randomly (frame by frame) from a pool of untreated, normally developing embryos. This comparison of a single test image with several reference images returned ten similarity profiles, in which the similarities of the test embryo to different developmental stages of reference embryos were displayed. The developmental stage of the test embryo was predicted by taking the timepoint of reference embryos at which the maximum similarity with the test images was the highest.

In a second approach, instead of reference images of different embryos, earlier acquisition images of the same embryo were used for similarity calculation for each acquisition timepoint of one timeseries acquisition, resulting in $k - 1$ similarity values at each acquisition timepoint index $k$. Changes of developmental epochs were located at local maxima of changes in similarity values.

### Growth rate and apparent activation energy estimation
To estimate the growth rate for each temperature, first the estimated developmental age for an image timeseries of the evaluated embryos was calculated. The data of all embryos were pooled and fitted with a linear model using the RANSAC (RANdom SAmple Consensus) algorithm with a minimum sample number of 2,000 and a residual threshold of 2.0. Then the growth rate ($g$) was defined as the slope of the fitted model.

To estimate the relative activation energy ($E_a$), the Arrhenius equation[43] was used as follows:

$$g = Ae^{-E_a/RT}$$

$$\ln g = \frac{-E_a}{R}\frac{1}{T} + c$$

with the universal gas constant $R = 8.314\,\text{J}\,\text{K}^{-1}\,\text{mol}^{-1}$. By fitting a linear model to the data using RANSAC, the apparent activation energy was estimated; 99.99% confidence intervals were obtained using bootstrapping with 100 samples.

### Phenotypic comparison of embryos at the same developmental stage

For comparison of phenotypic characteristics of zebrafish embryos, similarities were calculated between one image of a test embryo from 0.5 to 25 hpf and reference embryos at different developmental timepoints. For calculation of similarity distributions at different acquisition timepoints, a batch of $n$ embryos was selected, and $n \times (n-1)$ similarities between all embryos were calculated. For each embryo, similarities to other embryos were averaged. Variability of phenotypic characteristics was derived from the distribution width of similarity values at different acquisition timepoints.

### Detection of aberrant phenotypes with Twin Networks

Two approaches for the early detection of abnormal development were implemented using Twin Networks. First, defects were assessed based on the variation of predicted embryonic stages. Developmental stages of several embryos were predicted at each acquisition timepoint of a timeseries experiment with the previously described approach. Maldeveloping embryos were identified if their predicted developmental stage did not correspond to the expected developmental stage at the respective acquisition timepoint and the predicted stages for other embryos of the same batch.

Second, embryonic phenotypes of several embryos within a batch were compared among each other for each timepoint in the timeseries experiment, as described in the previous section. For each embryo, average similarity values served as an index representing the similarity of the phenotype of each embryo to the average phenotype of the embryo batch; $z$-scores were calculated for each embryo based on the mean and s.d. of the similarity indices within the respective embryo batch. In parallel, for each new acquisition timepoint in the timeseries experiment, the cumulative sum of the similarity indices for all previous acquisition timepoints was calculated individually for each embryo. Similar to the calculation of $z$-scores based on similarity indices calculated for a specific timepoint, $z$-scores were calculated for cumulative similarity indices for each acquisition timepoint of each embryo. Detection of deviation of embryonic phenotypes was performed based on the $z$-scores of both the similarity index calculated for the tested timepoint and the cumulative similarity index of each embryo.

### Detection of group phenotypes with Twin Networks

To identify drug-induced embryonic phenotypes, groups of embryos were compared with a reference group of untreated normally developing embryos. For each embryo, the similarity distance to the reference group was estimated by calculating the median of the similarity matrices obtained by comparing the test embryo series with each embryo series of the reference group. Next, the temporal series of similarity distributions was calculated for the reference group and the group of embryos to be evaluated. To test for significant differences in the temporal series similarity distributions between the reference and the test group, the nonparametric one-sided Mann–Whitney $U$ test over each timepoint of the image series was used. A threshold

$P$ value of 0.01 was applied to define significant differences. Then, a group of embryos was set to be detected as abnormal if a certain percentage of image frames were significantly different from the reference group. A fraction equal to 0.3 was used to define a detection; in other words, a set of embryos should be different from the reference group by at least 30% of the total imaging time to be considered as an abnormal detection.

The dependence of the accuracy of abnormality detection for different conditions with respect to the number of embryos used for the detection was evaluated as follows: first, a defined number of embryos for the test and reference groups was selected randomly from a pool of available ones (44, 65, 51, 50, 14, 47, 18 and 46 embryos for untreated, −BMP, −FGF, −Nodal, −PCP, −Shh, −Wnt and +RA embryos, respectively). Then, the groups of embryos were compared statistically as described above to determine whether the test group was detected as normal or abnormal. The process was done for 20 random samples of 3–44 embryos, and repeated five times. In the case of the detailed analysis of −BMP embryos, a pool of 79 (C5), 79 (C4), 117 (C3) and 88 (C2) embryos was used in addition to 17 embryos for the *bmp2b*-defective *swirl* mutant.

### Automatic generation of staging atlases from cosine similarities

Cosine similarity matrices were stored as .mat files exported from the Twin Network analysis, and subsequent results were stored as JSON files. A threshold was derived from the histogram of cosine similarity distributions. This threshold was used to mask areas of high noise, and values below that threshold were set to zero. Boundaries within the inverse of the sums of diagonals were identified as local maxima with find_peaks (scipy.signal, Scipy v.1.10.1). The first and last frames of the image sequence were set as additional boundaries. From the full set of embryos, sequences with comparable normal development were considered representative.

### Analysis of technical and biological variability in self-similarity matrices

Ten models of TwinNet were trained on a set of training images (61 embryos). All models were trained with the same embryo images and parameters, but the random image triplets and initial weights were different. Ten self-similarity matrices were then calculated for each embryo with one prediction per model. The variability arising from random variations in the models was assessed by analyzing the matrices generated by the different models for the same embryos (that is, mean and s.d.). For each embryo, an ensemble matrix (average similarity matrix) was calculated.

### Image processing for representative embryos in display items

Brightness and contrast in representative embryos were uniformly adjusted, and embryos were cropped manually in Fiji (ImageJ v.1.54f)[80], Adobe Illustrator (V. 26.2.1) and Adobe Photoshop (V. 23.3.1.426) along the chorion outlines to enhance visibility. Note that for illustration purposes a subset of embryo images was reused for display in different figures. Raw data are available from https://doi.org/10.48606/50.

### Ethics statement

All procedures involving animals were executed in accordance with the guidelines of the EU directive 2010/63/EU and the German Animal Welfare Act as approved by the local authorities represented by the Regierungspräsidium Tübingen and the Regierungspräsidium Freiburg. Experiments were performed exclusively with embryos and larvae that were not yet freely feeding.

### Reporting summary

Further information on research design is available in the Nature Portfolio Reporting Summary linked to this article.

## Data availability

Training, evaluation and temperature datasets are available from https://doi.org/10.48606/50. Additional data used for training and evaluation is available from https://doi.org/10.48606/15, https://www.youtube.com/watch?v=M2ApXHhYbaw (accessed on 20 March 2023) and https://doi.org/10.7554/eLife.07410.021. Source data are provided with this paper.

## Code availability

The Twin Network open-source code is available from https://github.com/mueller-lab/TwinNet.git (https://doi.org/10.5281/zenodo.8419446).

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

## Acknowledgements

We thank M. Dressler and A. A. Hyman for the permission to use their *C. elegans* data at https://www.youtube.com/watch?v=M2ApXHhYbaw. P.M. acknowledges funding from the European Research Council (ERC) under the European Union's Horizon 2020 research and innovation program (grant agreement No. 863952 (ACE-OF-SPACE)), the Max Planck Society, the EMBO Young Investigator Program, and the Deutsche Forschungsgemeinschaft (DFG, German Research Foundation) under Germany's Excellence Strategy—EXC 2117—422037984. This project has also received funding from the IZKF of the Medical Faculty of the University of Tübingen (to N.T. and P.M.). We are grateful to support from the Blue Sky research program of the University of Konstanz (Project EvoDevoGPT to P.M.).

## Author contributions

N.T., H.M.-N., M.Ü. and P.M. conceived the study. N.T., H.M.-N., M.Ü. and P.M. developed the methodology. N.T., H.M.-N., D.Č., J.G. and M.Ü. performed the investigation. N.T., H.M.-N., M.Ü. and P.M. visualized the data. P.M. acquired funding. P.M. was the project administrator. N.T., H.M.-N., D.Č., M.Ü. and P.M. wrote the manuscript.

## Competing interests

The authors declare no competing interests.

## Additional information

**Extended data** is available for this paper at https://doi.org/10.1038/s41592-023-02083-8.

**Correspondence and requests for materials** should be addressed to Murat Ünalan or Patrick Müller.

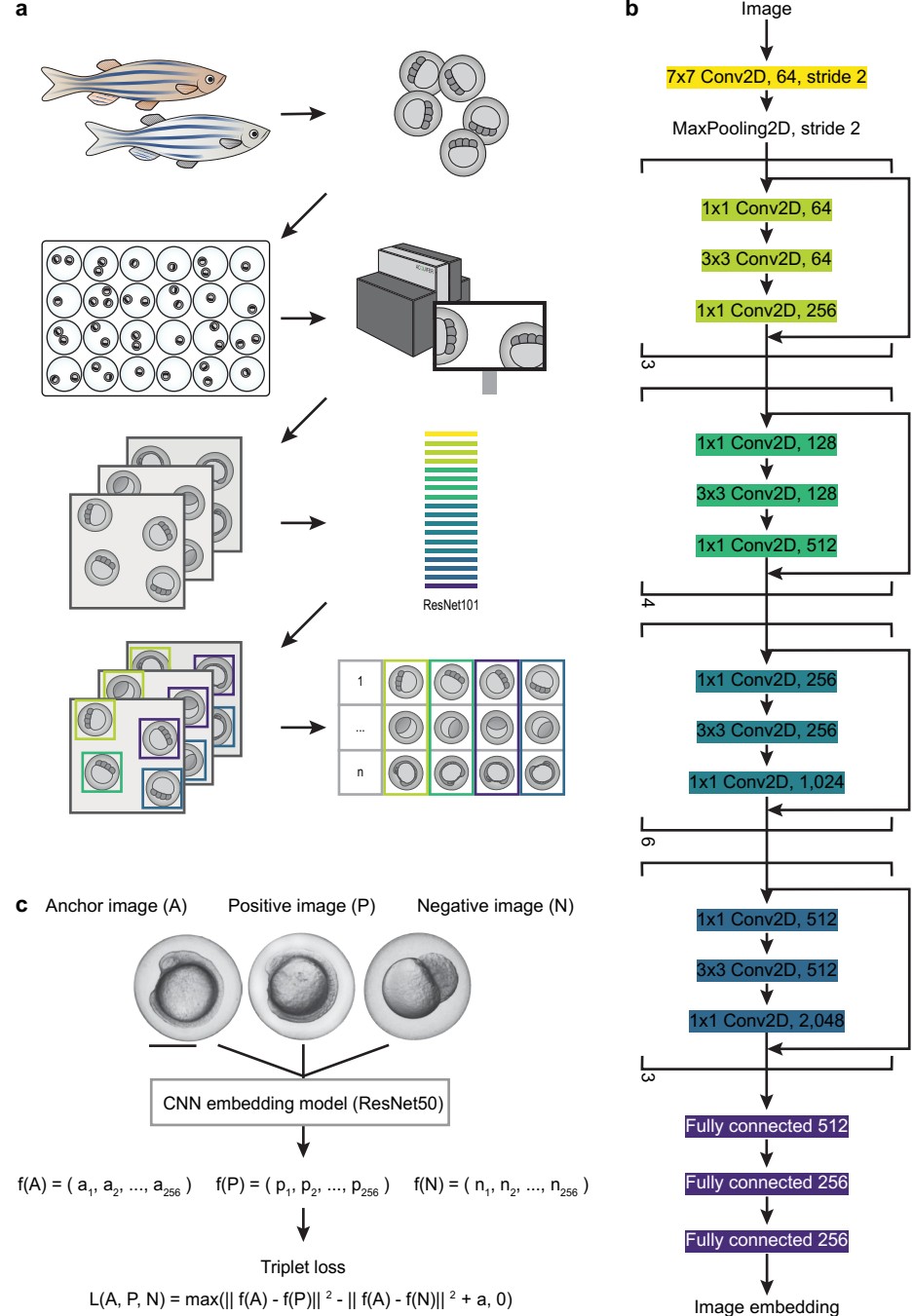

**a**

**b** Image

7x7 Conv2D, 64, stride 2

MaxPooling2D, stride 2

3 ⎰ 1x1 Conv2D, 64 / 3x3 Conv2D, 64 / 1x1 Conv2D, 256

4 ⎰ 1x1 Conv2D, 128 / 3x3 Conv2D, 128 / 1x1 Conv2D, 512

6 ⎰ 1x1 Conv2D, 256 / 3x3 Conv2D, 256 / 1x1 Conv2D, 1,024

3 ⎰ 1x1 Conv2D, 512 / 3x3 Conv2D, 512 / 1x1 Conv2D, 2,048

Fully connected 512

Fully connected 256

Fully connected 256

Image embedding

ResNet101

**c** Anchor image (A)　Positive image (P)　Negative image (N)

CNN embedding model (ResNet50)

$f(A) = (a_1, a_2, ..., a_{256})$　$f(P) = (p_1, p_2, ..., p_{256})$　$f(N) = (n_1, n_2, ..., n_{256})$

Triplet loss

$$L(A, P, N) = \max(\| f(A) - f(P)\|^2 - \| f(A) - f(N)\|^2 + a, 0)$$

**Extended Data Fig. 1 | Architecture of the Twin Network to analyze developmental dynamics. (a)** High-throughput imaging pipeline and ResNet101-based image segmentation to generate developmental trajectories of individual embryos. Embryos are individually tracked, as indicated by equally colored bounding boxes in the segmentation steps. **(b)** Model architecture of the core of the Twin Network based on ResNet50. **(c)** Image triplets consist of an anchor image, a positive image, and a negative image, and are passed to Twin Network for training with triplet loss. Anchor and positive images contain similar objects, while anchor and negative images show dissimilar objects. Triplet loss is used during the training to reduce the Euclidian distance between embeddings generated for the anchor image and positive image, and increase the distance between embeddings of the anchor image and negative image. Embryos for illustration also shown in Fig. 1a. Scale bar, 500 µm.

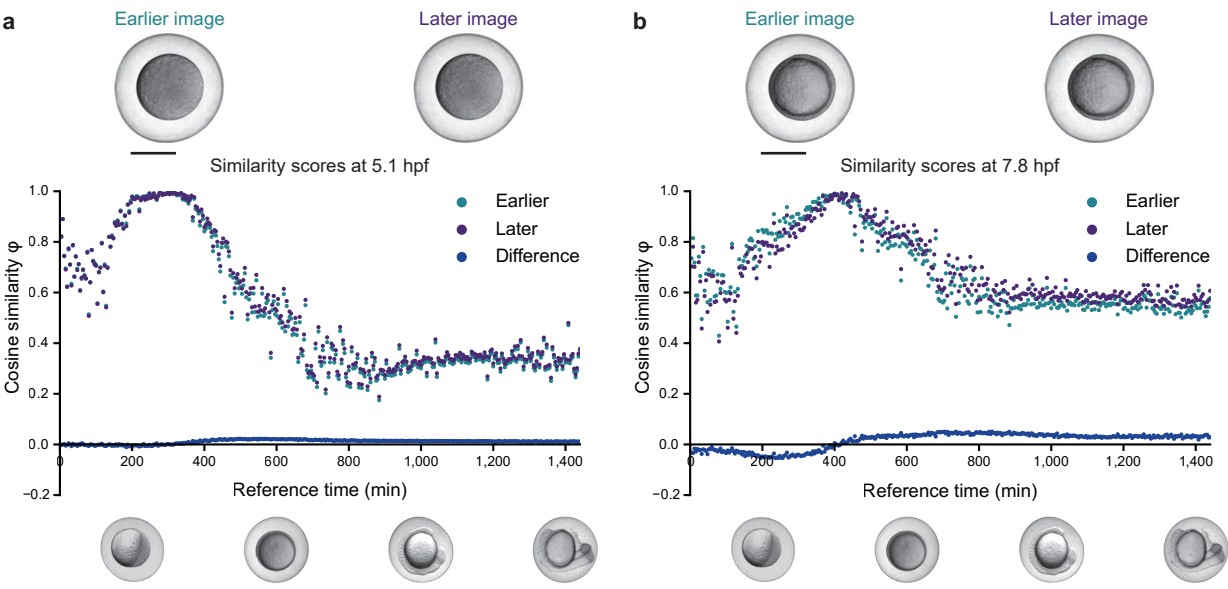

**Extended Data Fig. 2 | Identifying developmental progression with Twin Networks.** Comparison of two subsequently acquired images of the same embryo. Similarity plots calculated by comparison with reference images differ minimally with respect to the peak of similarity as well as similarity to distant embryonic stages. Subtracting the similarity of the earlier acquired image (turquoise) from the similarity profile of the later acquired image (purple) shows positivity following the peak of the similarity (blue), suggesting the attribution of the later acquired image towards later developmental stages. Comparisons of subsequent images taken 3 min and 8 seconds apart are shown for images of embryos captured at 5.1 hpf in (**a**) and of subsequent images of embryos captured at 7.8 hpf in (**b**). Two images per plot from the acquisition of one embryo, representative for three independent experiments, are shown. The temporal limits for this approach have not yet been defined. Embryos for illustration are also shown in Fig. 1c,e. Scale bars, 500 μm.

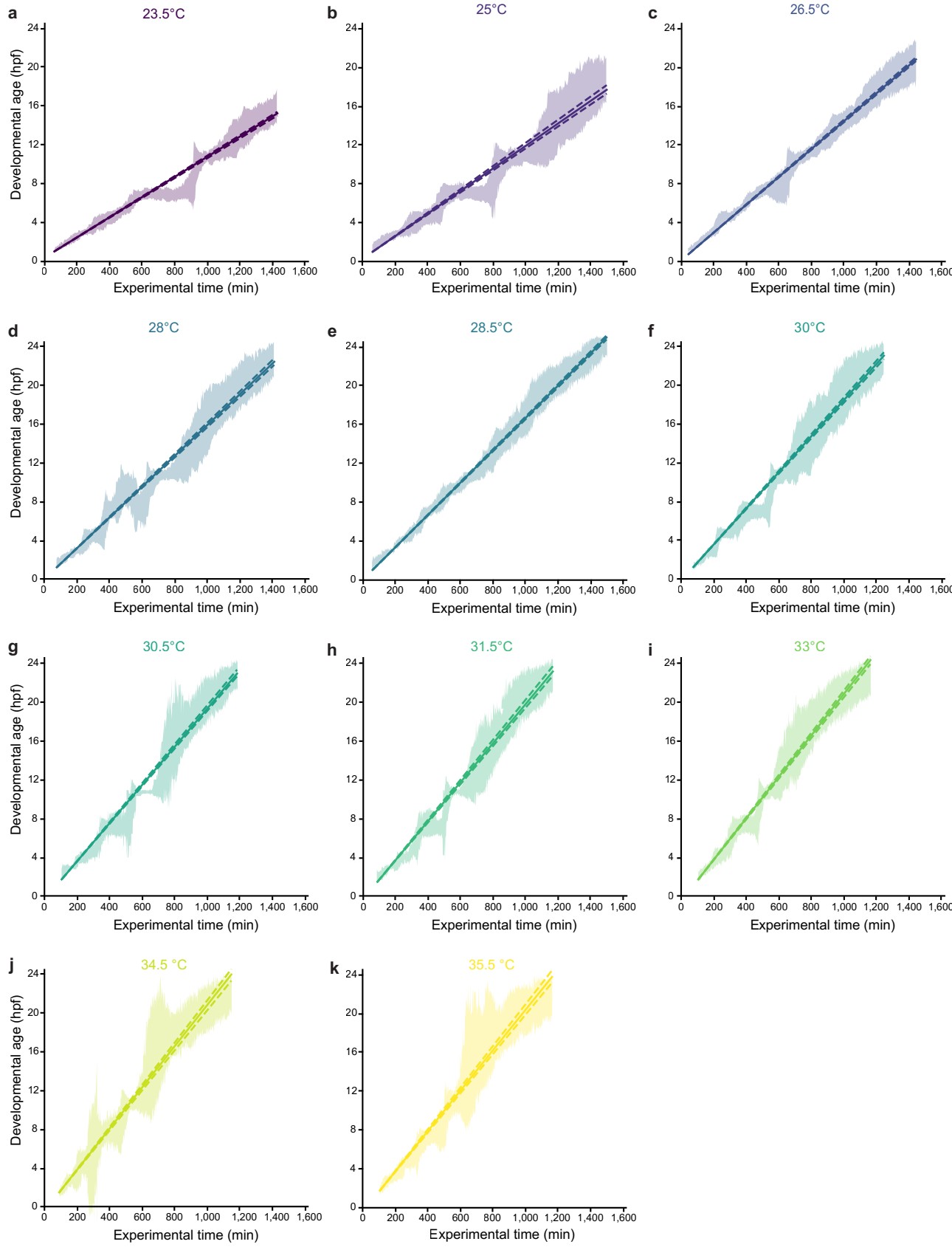

**Extended Data Fig. 3 | See next page for caption.**

**Extended Data Fig. 3 | Developmental age estimation for zebrafish embryos at different temperatures.** Error envelopes represent two times the median absolute deviation (MAD) and are shown together with the corresponding linear fit (solid line) plus 99% confidence interval (dashed lines). **(a)** 23.5 °C (slope = 0.623 (0.613, 0.630), $R^2$ = 0.972), **(b)** 25 °C (slope = 0.696 (0.680, 0.716), $R^2$ = 0.968), **(c)** 26.5 °C (slope = 0.861 (0.854, 0.871), $R^2$ = 0.982), **(d)** 28 °C (slope = 0.951 (0.937, 0.967), $R^2$ = 0.979), **(e)** 28.5 °C (slope = 1.000 (0.991, 1.009), $R^2$ = 0.987), **(f)** 30 °C (slope = 1.117, (1.099, 1.134), $R^2$ = 0.983), **(g)** 30.5 °C (slope = 1.182 (1.165, 1.204), $R^2$ = 0.981), **(h)** 31.5 °C (slope = 1.207 (1.183, 1.237), $R^2$ = 0.981), **(i)** 33 °C (1.284 (1.259, 1.305), $R^2$ = 0.983), **(j)** 34.5 °C (slope = 1.273 (1.238, 1.306), $R^2$ = 0.981), **(k)** 35.5 °C (slope = 1.246 (1.216, 1.287), $R^2$ = 0.981). n(23.5 °C) = 211, n(25 °C) = 198, n(26.5 °C) = 209, n(28 °C) = 168, n(28.5 °C) = 126, n(30 °C) = 187, n(30.5 °C) = 102, n(31.5 °C) = 130, n(33 °C) = 98, n(34.5 °C) = 70, n(35.5 °C) = 119. Data for 26.5 °C, 28.5 °C and 31.5 °C also shown in Fig. 2b.

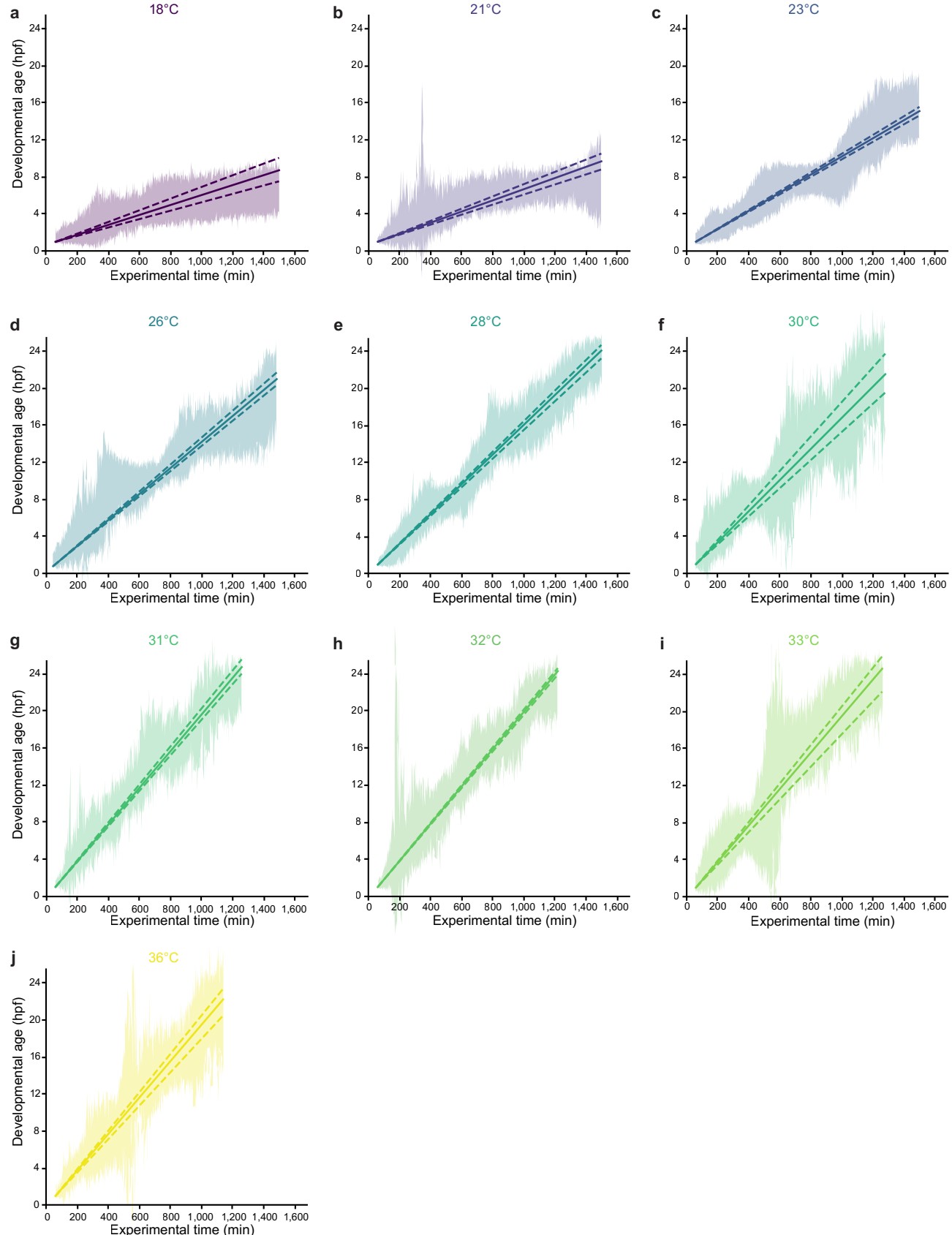

**Extended Data Fig. 4 | See next page for caption.**

**Extended Data Fig. 4 | Developmental age estimation for medaka embryos at different temperatures.** Error envelopes represent two times the median absolute deviation (MAD) and are shown together with the corresponding linear fit (solid line) plus 99% confidence intervals (dashed lines). **(a)** 18 °C (slope = 0.323 (0.272, 0.378), $R^2$ = 0.798), **(b)** 21 °C (slope = 0.361 (0.325, 0.397), $R^2$ = 0.825), **(c)** 23 °C (slope = 0.588 (0.568, 0.607), $R^2$ = 0.945), **(d)** 26 °C (slope = 0.842 (0.815, 0.872), $R^2$ = 0.965), **(e)** 28 °C (slope = 0.963 (0.929, 0.989), $R^2$ = 0.979), **(f)** 30 °C (slope = 0.966 (0.913, 1.119), $R^2$ = 0.966), **(g)** 31 °C (slope = 1.189 (1.153, 1.230), $R^2$ = 0.979), **(h)** 32 °C (slope = 1.207 (1.188, 1.224), $R^2$ = 0.978), **(i)** 33 °C (slope = 1.175 (1.059, 1.250), $R^2$ = 0.980), **(j)** 36 °C (slope = 1.180 (1.084, 1.245), $R^2$ = 0.974). n(18 °C) = 65, n(21 °C) = 32, n(23 °C) = 92, n(26 °C) = 47, n(28 °C) = 46, n(30 °C) = 41, n(31 °C) = 21, n(32 °C) = 40, n(33 °C) = 42, n(36 °C) = 35. Data for 26 °C, 28 °C and 31 °C also shown in Fig. 2e.

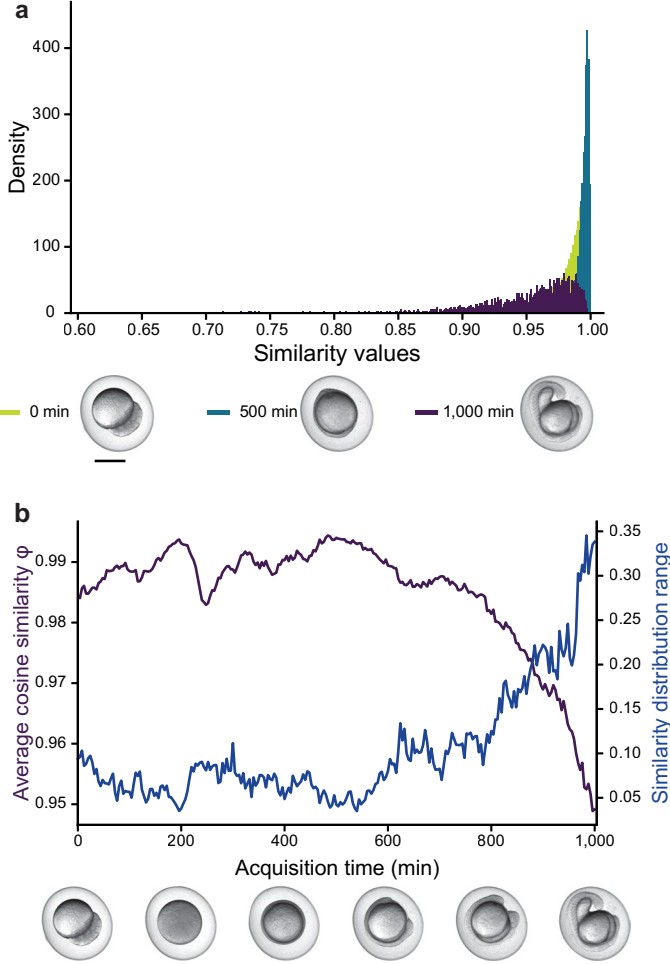

**Extended Data Fig. 5 | Characterization of morphological variability during zebrafish development. (a)** Distribution widths of similarity values at different acquisition time points calculated for 77 embryos. The distribution width of similarities is wider at later acquisition time points. **(b)** Relation of average similarities (purple) and distribution width of similarity values (blue) at different embryonic stages. Representative images of embryos at corresponding developmental stages are shown below the x-axis. Average similarity is constant until gastrulation and then decreases. Distribution width of similarities is low until gastrulation and increases steplike after gastrulation. The embryo images are representative examples of the whole sample group (n = 77). Scale bars, 500 μm.

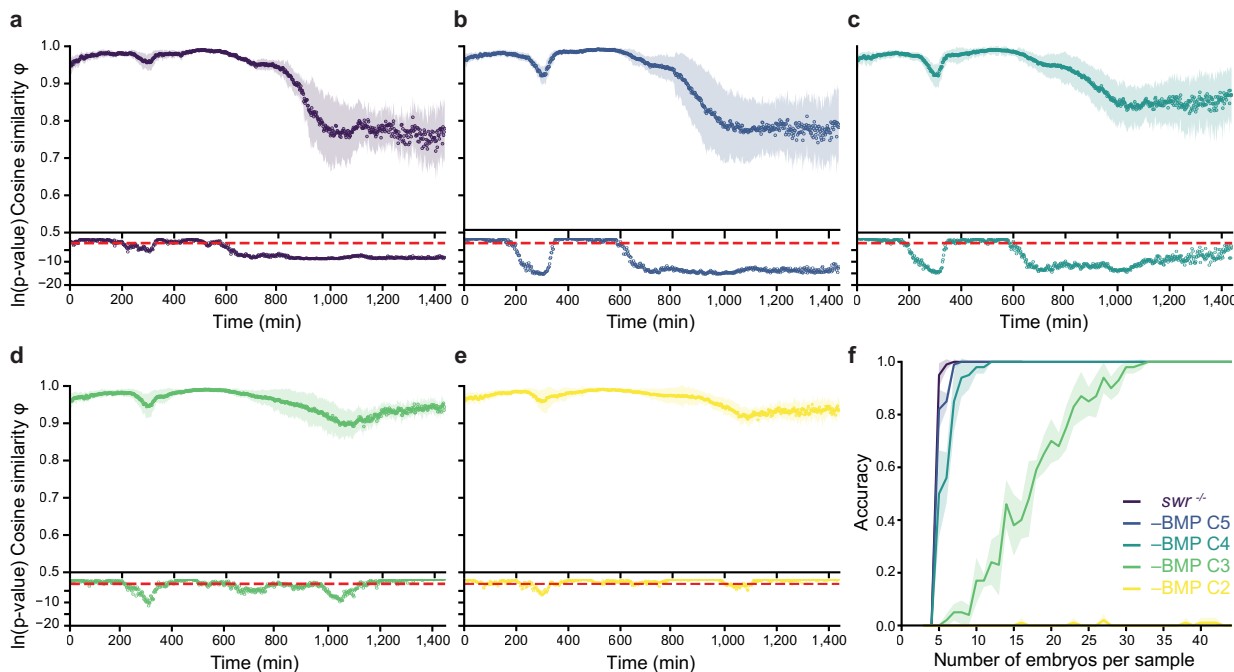

**Extended Data Fig. 6 | Distinguishing phenotypes of different severity during zebrafish development. (a-f)** Detection of -BMP phenotypes of different strength. (**a-e**) Upper panels show the mean similarities and standard deviation of similarities of *bmp* (*swr^-/-*) mutants (a) and -BMP drug-treated embryos with C5 (b), C4 (c), C3 (d) and C2 (e) phenotypes. The respective lower panels show significance levels of the difference from untreated embryos along the time axis in p-values determined using a nonparametric one-sided Mann-Whitney U test over each time point of the image series. No adjustments for multiple comparisons were made. n = 44 for all cases. (**f**) Dependency of the accuracy of abnormality detection on the number of embryos used for the analysis. Mean and standard deviation are shown for five repetitions with randomly selected samples. Raw data for analysis from https://doi.org/10.48606/15.

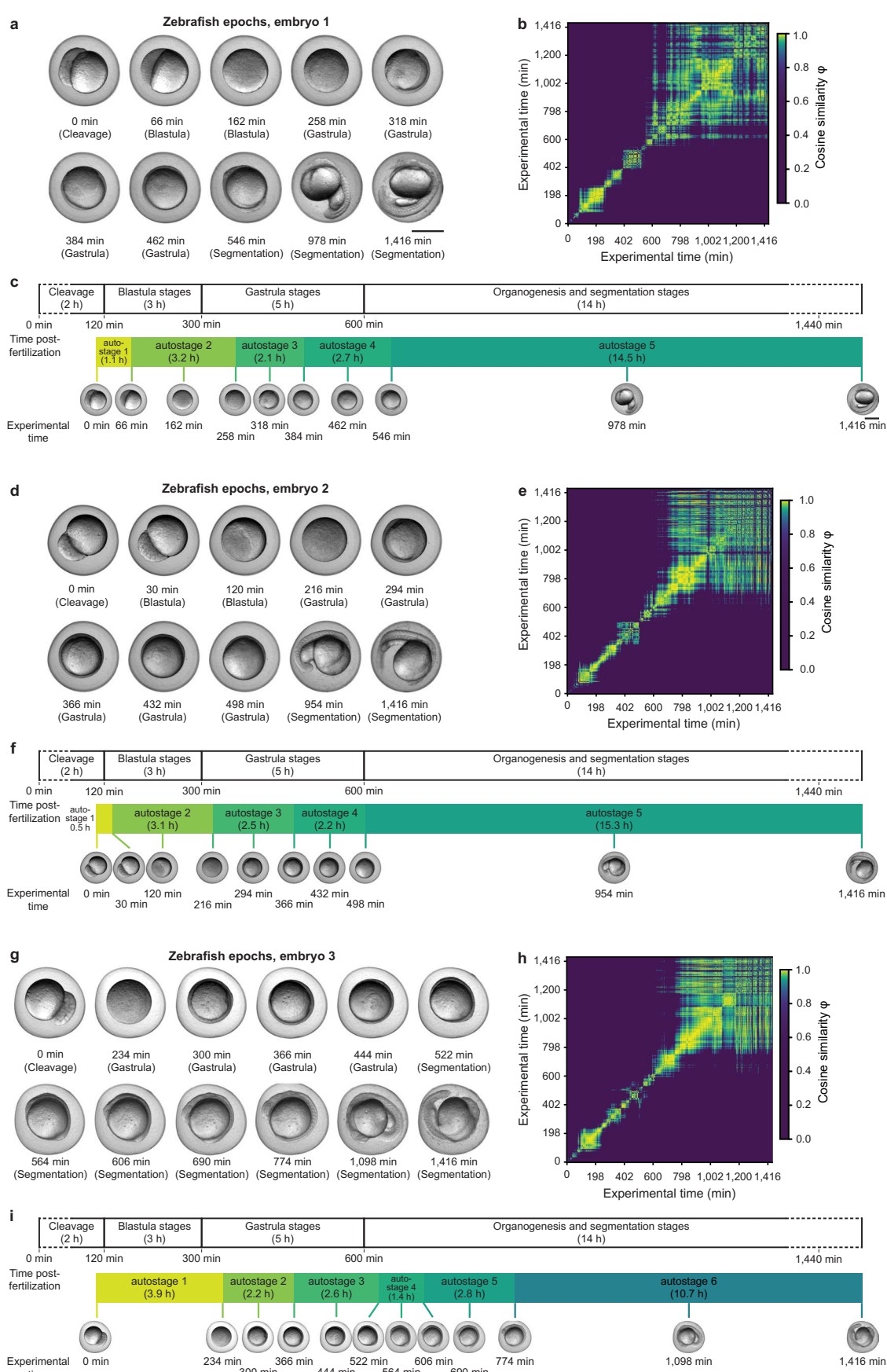

**Extended Data Fig. 7 | See next page for caption.**

**Extended Data Fig. 7 | Automatic detection of developmental epochs and transitions in zebrafish (*Danio rerio*) embryos.** The Twin Network detects and partitions embryo development into phases that are in line with the classical zebrafish staging atlas[6]. The term *autostage* describes a time phase within the recorded developmental period of an embryo that can be delineated by a plateau of coherently high similarity values calculated using the Twin Network. These similarity values were calculated by self-similarity comparison with images of previous developmental stages of the same test embryo. **(a)** Automatically selected images at the beginning, in the middle, and at the end of Twin Network-predicted plateaus of similarity values, that is autostages, for one test zebrafish embryo (embryo 1). Embryos for illustration in (a) also shown in Fig. 5b. **(b)** Calculated similarities used as the basis for the selection of depicted images of embryo 1. **(c)** Time points in the classical staging atlas[6] are shown at the top. Automatically generated autostages were calculated based on phases of high similarity in embryo morphology and are shown below. Embryos for illustration in (c) also shown in Fig. 5b. **(d)** Automatically selected images based on autostages as described in (a) for a zebrafish embryo (embryo 2). **(e)** Calculated similarities used as the basis for the selection of the depicted images of embryo 2. **(f)** Time points in the classical staging atlas[6] are shown at the top. Autostages were calculated based on phases of high similarity in embryo morphology and are shown below. **(g)** Automatically selected images based on autostages as described in (a) for a zebrafish embryo (embryo 3). **(h)** Calculated similarities used as the basis for the selection of the depicted images of embryo 3. **(i)** Time points in the classical staging atlas[6] are shown at the top. Autostages were calculated based on phases of high similarity in embryo morphology and are shown below; n = 3 out of 131 representative embryos. Images in (a), (d) and (g) correspond to the pictograms in (c), (f) and (i) at the indicated timepoints. The reference stages in the upper panels of (c), (f) and (i) are annotated with the time postfertilization (min). The example images in (a), (d) and (g), the similarities in (b), (e) and (h), and the autostages in the lower panels of (c), (f) and (i) are annotated with the experimental time (min). Imaging was started at 2 hpf (64-cell stage). Scale bars, 500 µm.

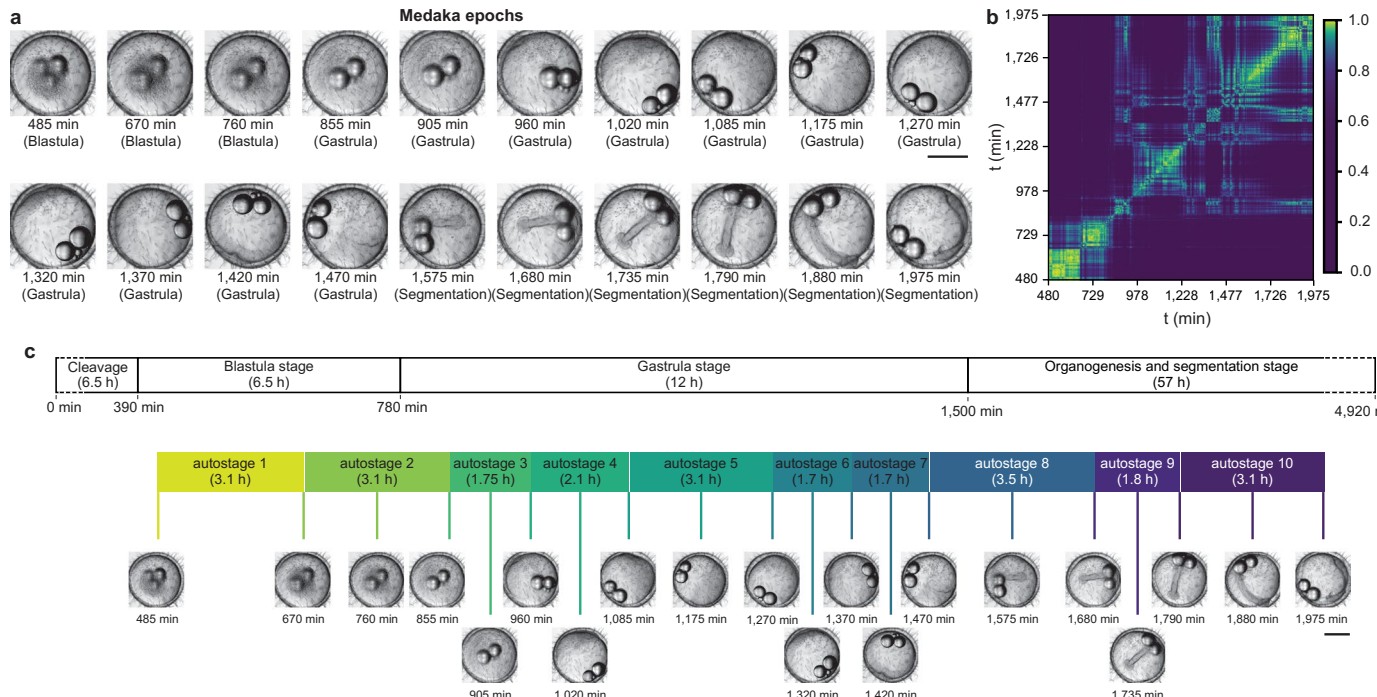

**Extended Data Fig. 8 | Automatic detection of developmental epochs and transitions in medaka (*Oryzias latipes*) embryos.** The Twin Network detects and partitions embryo development into phases that are in line with the classical medaka staging atlas[7]. **(a)** Automatically selected images from autostages and boundaries for a single embryo. **(b)** Calculated similarities used as the basis for the selection of the depicted images. **(c)** Time points in the classical medaka staging atlas are shown at the top. Automatically generated autostages were calculated based on phases of high similarity in embryo morphology and are shown below; n = 1 representative out of 232 embryos. Images in (a) correspond to the pictograms in (c) at the indicated timepoints. Scale bars, 500 μm. Raw data for analysis from https://doi.org/10.48606/15.

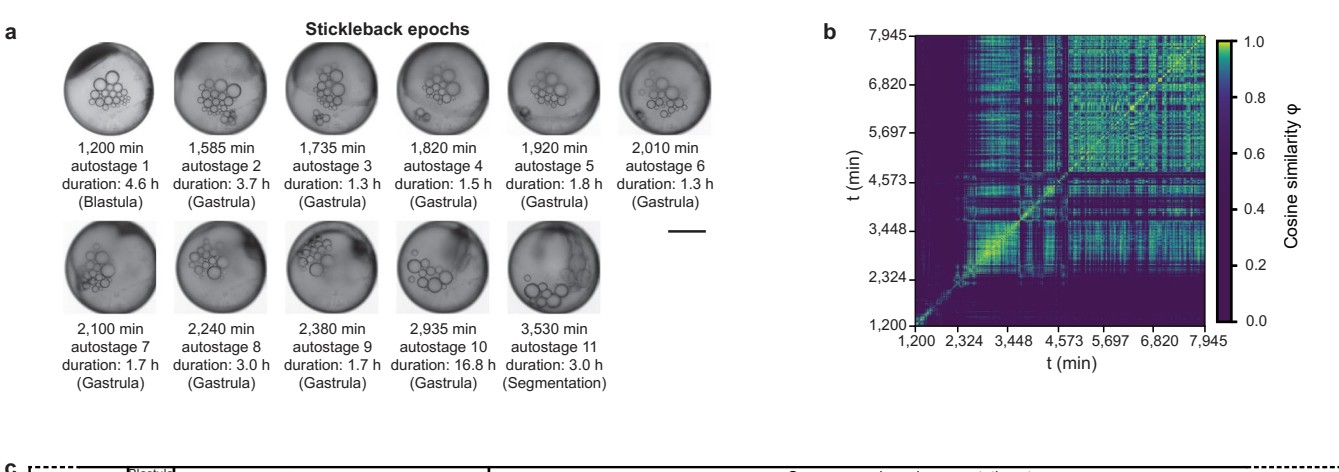

**Extended Data Fig. 9 | Automatic detection of developmental epochs and transitions in three-spined stickleback (*Gasterosteus aculeatus*) embryos.** The Twin Network detects and partitions embryo development into phases that are in line with the classical stickleback staging atlas[9]. **(a)** A selected set of images from autostages and boundaries for a single embryo. **(b)** Calculated similarities used as the basis for the depicted images. **(c)** Time points in the classical staging atlas[9] are shown at the top. Autostages were calculated based on phases of high similarity in embryo morphology; n = 1 representative out of 56 embryos. Images in (a) correspond to the pictograms in (c) at the indicated timepoints. Scale bars, 500 μm. Raw data for analysis from https://doi.org/10.48606/15.

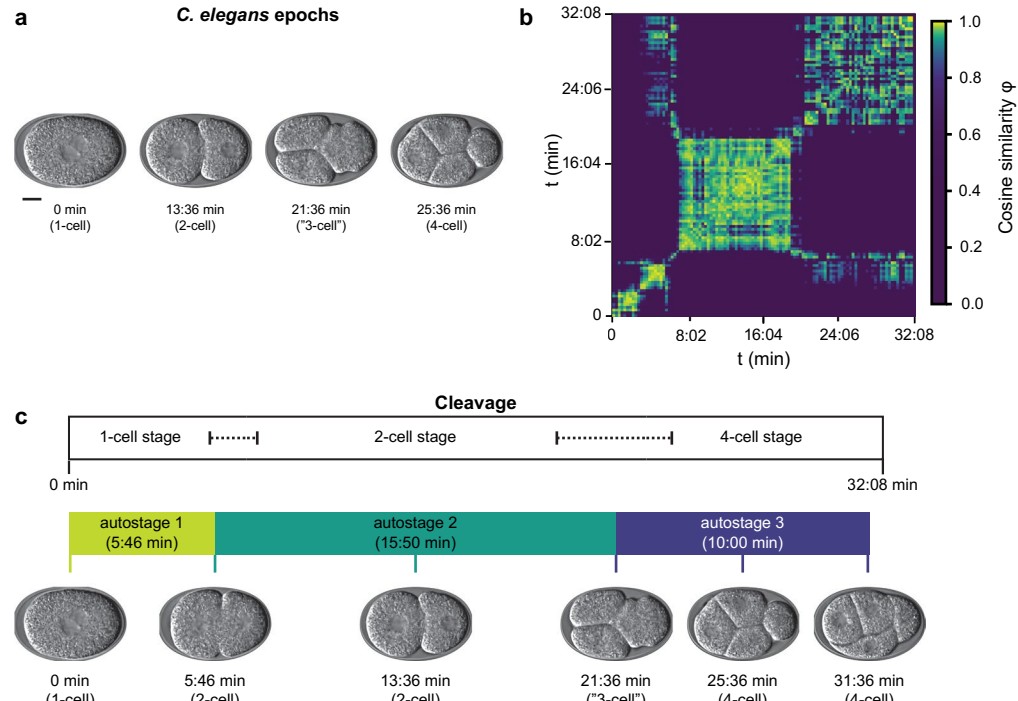

**Extended Data Fig. 10 | Automatic detection of cell divisions in nematode (*Caenorhabditis elegans*) embryos.** The Twin Network detects and partitions development into phases that are in line with human staging and early embryogenesis descriptions (http://www.wormbook.org)[14]. **(a)** A selected set of images from autostages and boundaries for a single embryo. **(b)** Calculated similarities used as the basis for the depicted images. Gross homology to a distant morphology at 5–7 min is present around 20–32 min of the acquisition.

**(c)** Dashed lines indicate cytokinesis phases as detected by a human observing the original video. Automatically generated autostages were calculated based on phases of high similarity in embryo morphology and are shown below (frames taken in intervals of 17.5 s). Notably, the blastomere divisions giving rise to ABa, ABp, EMS and P2 cells were correctly identified; n = 1 embryo. Images in (a) correspond to the pictograms in (c) at the indicated timepoints. Scale bar, 10 μm. Raw data from https://www.youtube.com/watch?v=M2ApXHhYbaw.

# Reporting Summary

## Statistics

For all statistical analyses, confirm that the following items are present in the figure legend, table legend, main text, or Methods section.

| n/a | Confirmed | |
|---|---|---|
| ☐ | ☒ | The exact sample size (*n*) for each experimental group/condition, given as a discrete number and unit of measurement |
| ☐ | ☒ | A statement on whether measurements were taken from distinct samples or whether the same sample was measured repeatedly |
| ☐ | ☒ | The statistical test(s) used AND whether they are one- or two-sided *Only common tests should be described solely by name; describe more complex techniques in the Methods section.* |
| ☒ | ☐ | A description of all covariates tested |
| ☐ | ☒ | A description of any assumptions or corrections, such as tests of normality and adjustment for multiple comparisons |
| ☐ | ☒ | A full description of the statistical parameters including central tendency (e.g. means) or other basic estimates (e.g. regression coefficient) AND variation (e.g. standard deviation) or associated estimates of uncertainty (e.g. confidence intervals) |
| ☐ | ☒ | For null hypothesis testing, the test statistic (e.g. *F*, *t*, *r*) with confidence intervals, effect sizes, degrees of freedom and *P* value noted *Give P values as exact values whenever suitable.* |
| ☒ | ☐ | For Bayesian analysis, information on the choice of priors and Markov chain Monte Carlo settings |
| ☒ | ☐ | For hierarchical and complex designs, identification of the appropriate level for tests and full reporting of outcomes |
| ☒ | ☐ | Estimates of effect sizes (e.g. Cohen's *d*, Pearson's *r*), indicating how they were calculated |

*Our web collection on statistics for biologists contains articles on many of the points above.*

## Software and code

Policy information about availability of computer code

| Data collection | For data acquisition on an Acquifer Imaging Machine we used the Imaging Machine control software (Acquifer Imaging GmbH, Version ID 4.00.21). Additionally, two Keyence BZ-X810 microscopes with the BZ-X800 viewer (Keyence, Version 01.03.00.01) were used. One of the Keyence BZ-X810 microscopes was equipped with a stage-top incubator (Oko-lab H301-KEYENCE-BZX with a UNO temperature controller). |
|---|---|
| Data analysis | For image annotation, we used the Visual Object Tagging Tool (Microsoft, https://github.com/microsoft/VoTT, Version 2.2.0). For model training and testing, we used custom Twin Network software (https://github.com/mueller-lab/TwinNet). The training was performed on NVIDIA RTX 3070 and 3090 cards (ASUS) on Windows 10 and Ubuntu 20.04, CUDA 11.2. For comparison to vector diffusion maps, we used software provided with the original publication (Dsilva et al., Development 2012) in MATLAB R2022a (MathWorks). Segmented time-series of single embryos were loaded into Fiji (ImageJ 1.54f), Adobe Illustrator (V. 26.2.1) and Adobe Photoshop (V. 23.3.1.426) for visualization and manual cropping. |

For manuscripts utilizing custom algorithms or software that are central to the research but not yet described in published literature, software must be made available to editors and reviewers. We strongly encourage code deposition in a community repository (e.g. GitHub). See the Nature Portfolio guidelines for submitting code & software for further information.

## Data

Policy information about availability of data

All manuscripts must include a data availability statement. This statement should provide the following information, where applicable:

- Accession codes, unique identifiers, or web links for publicly available datasets
- A description of any restrictions on data availability
- For clinical datasets or third party data, please ensure that the statement adheres to our policy

Training, evaluation and temperature data sets are available from https://doi.org/10.48606/50. Additional data used for training and evaluation are available from https://doi.org/10.48606/15, https://www.youtube.com/watch?v=M2ApXHhYbaw (accessed on 03/20/2023) and https://doi.org/10.7554/eLife.07410.021. Source data for all graphs is provided in separate source data files alongside this paper.

## Human research participants

Policy information about studies involving human research participants and Sex and Gender in Research.

| | |
|---|---|
| Reporting on sex and gender | N/A |
| Population characteristics | N/A |
| Recruitment | N/A |
| Ethics oversight | N/A |

Note that full information on the approval of the study protocol must also be provided in the manuscript.

# Field-specific reporting

Please select the one below that is the best fit for your research. If you are not sure, read the appropriate sections before making your selection.

☒ Life sciences ☐ Behavioural & social sciences ☐ Ecological, evolutionary & environmental sciences

For a reference copy of the document with all sections, see nature.com/documents/nr-reporting-summary-flat.pdf

# Life sciences study design

All studies must disclose on these points even when the disclosure is negative.

| | |
|---|---|
| Sample size | To determine a suitable sample size for the development of Twin Network as well as the segmentation network, we used an active learning approach. In an iterative process, we progressively increased the number of images/embryos used as training and validation sets until training metrics on the validation set reached a saturation level. The pool out of which the images were selected comprised more than 15,000 embryos. For data analysis sample sizes of at least 5 embryos were found to provide good accuracy (Fig. 4j and Extended Data Fig. 6f). Analyses of morphological variability and variability of predicted embryonic stages between normally developing embryos were performed on 77 embryos acquired in one experiment. Comparisons of morphological differences between normally and abnormally developing embryos were performed between 1 maldeveloping and 6 normally developing embryos from one image acquisition. Differences of predicted stages for normally and abnormally developing embryos were shown for 14 maldeveloping and 7 normally developing embryos from one experiment. Autostaging was performed on 131 zebrafish, 56 stickleback, 232 medaka and one C. elegans embryo. For the temperature analysis, 61 zebrafish and 146 medaka embryos were used for training. The zebrafish test data sample size per temperature was: n(23.5°C) = 211, n(25°C) = 198, n(26.5°C) = 209, n(28°C) = 168, n(28.5°C) = 126, n(30°C) = 187, n(30.5°C) = 102, n(31.5°C) = 130, n(33°C) = 98, n(34.5°C) = 70, n(35.5°C) = 119; and for medaka: n(18°C) = 65, n(21°C) = 32, n(23°C) = 92, n(26°C) = 47, n(28°C) = 46, n(30°C) = 41, n(31°C) = 21, n(32°C) = 40, n(33°C) = 42, n(36°C) = 35. For the comparison to the vector diffusion maps approach (Dsilva et al., Development 2012), 2 zebrafish embryos were analyzed. |
| Data exclusions | Embryos were excluded from data sets based on one of several criteria:<br>- Partial visibility in images<br>- Microscope image was taken without the embryo being in the correct focal plane<br>- Embryo showed visible signs of artificially induced or natural malformation during embryogenesis<br>- Particles obstructed the view of parts of the embryo<br>- Illumination of the embryo during acquisition was insufficient or unbalanced<br>- Embryos were unfertilized<br>- Tracking was inconsistent, e.g. due to extensive embryonic movement |
| Replication | Microscopy experiments for the provided training and testing data sets of Twin Network were carried out 34 times, and data was collected reliably and with comparable quality. Instructions for the replication of analyses using Twin Network and corresponding follow-along scripts are provided at https://github.com/mueller-lab/TwinNet. For embryonic age assessment experiments, at least three biological replicates were performed. The test data collection for the temperature analysis was performed once per temperature. |

| Randomization | Embryos from each species were randomly allocated into experimental groups. |
| Blinding | Since embryos from each species were indistinguishable in different experiments, blinding of the investigators was not necessary. |

# Reporting for specific materials, systems and methods

We require information from authors about some types of materials, experimental systems and methods used in many studies. Here, indicate whether each material, system or method listed is relevant to your study. If you are not sure if a list item applies to your research, read the appropriate section before selecting a response.

## Materials & experimental systems

| n/a | Involved in the study |
|-----|-----------------------|
| ☒ | Antibodies |
| ☒ | Eukaryotic cell lines |
| ☒ | Palaeontology and archaeology |
| ☐ | ☒ Animals and other organisms |
| ☒ | Clinical data |
| ☒ | Dual use research of concern |

## Methods

| n/a | Involved in the study |
|-----|-----------------------|
| ☒ | ChIP-seq |
| ☒ | Flow cytometry |
| ☒ | MRI-based neuroimaging |

## Animals and other research organisms

Policy information about studies involving animals; ARRIVE guidelines recommended for reporting animal research, and Sex and Gender in Research

| Laboratory animals | We performed experiments exclusively on embryos and larvae that were not yet freely feeding. We used zebrafish of different genetic backgrounds to maximize the utility of the approach:<br>- Wild type TE (Pomreinke et al., eLife 2017)<br>- Tg(sebox:EGFP) (Poulain et al., Development 2002)<br>- Tg(gsc:GFP) (Doitsidou et al, Cell 2002)<br>- Tg(gsc:TurboRFP) (Sako et al., Cell Reports 2016)<br>- Tg(lhx1a:EGFP) (Swanhart et al., Int J Dev Biol 2010)<br>- sqt+/- (Dougan et al., Development 2003)<br>Age of embryos: 0-27 hpf.<br>In addition, medaka eggs of the Cab strain were used. |
| Wild animals | We did not use wild animals. |
| Reporting on sex | Sex-based analysis was not performed because phenotypical sex identification is not possible in early embryos. |
| Field-collected samples | We did not use field-collected samples. |
| Ethics oversight | All procedures involving animals were executed in accordance with the guidelines of the EU directive 2010/63/EU and the German Animal Welfare Act as approved by the local authorities represented by the Regierungspräsidium Tübingen and the Regierungspräsidium Freiburg. Experiments were executed exclusively on embryos and larvae that were not yet freely feeding. |

Note that full information on the approval of the study protocol must also be provided in the manuscript.

