## [Peer Review File · Nature Methods]

Peer Review Information

Manuscript Title: Uncovering developmental time and tempo using deep learning

Corresponding author name(s): Murat Ünalán, Patrick Müller

Editorial Notes: none

Reviewer Comments & Decisions:

Decision Letter, initial version:

Dear Patrick,

Your Article, "Uncovering developmental time and tempo using deep learning", has now been seen by 3 reviewers. As you will see from their comments below, although the reviewers find your work of considerable potential interest, they have raised a number of concerns. We are interested in the possibility of publishing your paper in Nature Methods, but would like to consider your response to these concerns before we reach a final decision on publication.

We therefore invite you to revise your manuscript to address these concerns. In particular, I recommend adding a strong biological demonstration to support the method. I also recommend adding discussion on whether this approach can be extended to analysis of other species.

* include a point-by-point response to the reviewers and to any editorial suggestions

* please underline/highlight any additions to the text or areas with other significant changes to facilitate review of the revised manuscript

- * address the points listed described below to conform to our open science requirements
- * ensure it complies with our general format requirements as set out in our guide to authors at www.nature.com/naturemethods
- * resubmit all the necessary files electronically by using the link below to access your home page

[Redacted] This URL links to your confidential home page and associated information about manuscripts you may have submitted, or that you are reviewing for us. If you wish to forward this email to co-authors, please delete the link to your homepage.

We hope to receive your revised paper within eight weeks. If you cannot send it within this time, please let us know. In this event, we will still be happy to reconsider your paper at a later date so long as nothing similar has been accepted for publication at Nature Methods or published elsewhere.

OPEN SCIENCE REQUIREMENTS

REPORTING SUMMARY AND EDITORIAL POLICY CHECKLISTS

Please note that these forms are dynamic 'smart pdfs' and must therefore be downloaded and completed in Adobe Reader. We will then flatten them for ease of use by the reviewers. If you would

like to reference the guidance text as you complete the template, please access these flattened versions at <http://www.nature.com/authors/policies/availability.html>.

DATA AVAILABILITY

We strongly encourage you to deposit all new data associated with the paper in a persistent repository where they can be freely and enduringly accessed. We recommend submitting the data to discipline-specific and community-recognized repositories; a list of repositories is provided here:

<http://www.nature.com/sdata/policies/repositories>

All novel DNA and RNA sequencing data, protein sequences, genetic polymorphisms, linked genotype and phenotype data, gene expression data, macromolecular structures, and proteomics data must be deposited in a publicly accessible database, and accession codes and associated hyperlinks must be provided in the “Data Availability” section.

Please include a “Data availability” subsection in the Online Methods. This section should inform readers about the availability of the data used to support the conclusions of your study, including accession codes to public repositories, references to source data that may be published alongside the paper, unique identifiers such as URLs to data repository entries, or data set DOIs, and any other statement about data availability. At a minimum, you should include the following statement: “The data that support the findings of this study are available from the corresponding author upon request”, describing

which data is available upon request and mentioning any restrictions on availability. If DOIs are provided, please include these in the Reference list (authors, title, publisher (repository name), identifier, year). For more guidance on how to write this section please see: <http://www.nature.com/authors/policies/data/data-availability-statements-data-citations.pdf>

CODE AVAILABILITY

Please include a “Code Availability” subsection in the Online Methods which details how your custom code is made available. Only in rare cases (where code is not central to the main conclusions of the paper) is the statement “available upon request” allowed (and reasons should be specified).

For more information on our code sharing policy and requirements, please see: <https://www.nature.com/nature-research/editorial-policies/reporting-standards#availability-of-computer-code>

MATERIALS AVAILABILITY

SUPPLEMENTARY PROTOCOL

To help facilitate reproducibility and uptake of your method, we ask you to prepare a step-by-step Supplementary Protocol for the method described in this paper. We [encourage authors to share their step-by-step experimental protocols](https://www.nature.com/nature-research/editorial-policies/reporting-standards#protocols) on a protocol sharing platform of their choice and report the protocol DOI in the reference list. Nature Portfolio's Protocol Exchange is a free-to-use and open resource for protocols; protocols deposited in Protocol Exchange are citable and can be linked from the published article. More details can found at [a](https://www.nature.com/nature-research/editorial-policies/reporting-standards#protocols)

href="https://www.nature.com/protocolexchange/about"
target="new">www.nature.com/protocolexchange/about.

ORCID

Sincerely,
Madhura

Madhura Mukhopadhyay, PhD
Senior Editor
Nature Methods

Reviewers' Comments:

Reviewer #1:

Remarks to the Author:

The authors present essentially a robust method to compute and quantify phenotypical differences between images of larvae. Application of this method to a huge number of images from developing zebrafish embryos, by comparing different stages, results in the definition of a development trajectory for normal embryos. This is then used to analyze variability between "normal" individuals during the first 24 hrs of development, to identify individuals that deviate from the normal trajectory, and to compare trajectories of normal embryos from that of embryos treated with inhibitors for BMP or Nodal signaling.

Finally, the authors show that comparison of images from a single embryo, but at different stages, allows for definition of developmental epochs and stages, which they also apply to less well characterized species such as medaka, stickleback, and even *C. elegans*.

This approach is certainly novel, it is convincing in the results it provides, and impressive in the scale of work presented. Only minor points need to be addressed:

Minor points:

- 1) In line 62, the authors state that their dataset comprises 3 million images, in Fig. 1a they state 15 million: how comes? Could it be that the 15 million were required to obtain the 3 million "high quality" images used for model training? What happened to the remaining one's, were they ever used?
- 2) Lines 152-175: the authors clearly illustrate that defects due to depletion of specific signaling pathways will be detected by their developmental staging method, however without giving any information about the specific defect that causes the divergence from the normal staging trajectory. Possibly (presumably), treatment with another compound affecting BMP signaling would result in a trajectory indistinguishable from the one shown here. However, I also presume that very different phenotypes may occur at similar stages, thus yielding a similar deviation from the normal trajectory, but also from the BMP-inhibition one. Would it be useful to have a "developmental trajectory database" that could be used for new studies, on other compounds even in a different lab. Could another lab use the images from this work and run the comparison with its own images, or would all the images need to be recaptured (different setting, different conditions)?
- 3) Supplementary Fig. 6: I am not sure this representation is adding to the comprehension, given that Fig. 4b is already quite clear.
- 4) Supplementary Fig. 8c: there seems to be a mismatch between the stage timing in minutes indicated along the images and the stage timing in hours in the "autostage strip".
- 5) How many images were used for the medaka, stickleback and *C. elegans* autostage determination? I finally found it in "Image acquisition", but it should be indicated maybe in the figure legends. A lower number of images may explain the plethora of autostages seen for stickleback, given the presumably also high variability that was also observed in zebrafish.

Marc Muller

Reviewer #2:

Remarks to the Author:

none This manuscript is a follow-up paper of a previously published study employing neural networks to phenotype embryogenesis in fish. The current work makes substantial advances and is conceptually different as it employs a twin-network architecture. This interesting methodology allows the authors to quantify morphological distances in an unbiased manner. It is applied to normal development and

signaling perturbation and even extended to other species of fish and even *C. elegans*. In general, the paper is informative, presents a novel approach and appears scientifically sound and well documented. The biological findings confirm long known concepts and are rather unsurprising, thus this is rather a methodological paper that does not provide much novel biological insight. However, the potential to build upon this methodological framework is large and thus, I think it is of exceptionally high scientific interest to a larger audience.

I have two main questions:

Fig. 3 depicts how phenotypic differences after signaling perturbation can be detected in an unbiased manner. That is cool and has a multitude of potential applications. However, I didn't understand how robust in terms of sensitivity or specificity such a phenotyping method would be. Given that the perturbation is known, can the parameter space in which this is feasible be defined more clearly? I would love to get a feeling for how many embryos need to be treated and how penetrant the phenotype would have to be to be distinguishable from random noise, and how and where the detection would reach its limits. I know this depends on multiple, investigator specific parameters, threshold levels, ... So I don't expect an easy answer, but could the dataset the authors gathered serve as an example case to derive a clearer picture of how usable the twin-network phenotyping would be in an applied setting? Is there a way to quantify how robust this method would distinguish a phenotype without prior knowledge? What parameters are crucial in which ranges?

The distance matrices across developmental stages are impressive. It seems that Fig. 4 And those shown in Suppl Fig. 7 were each derived from a single embryo. Interestingly and unsurprisingly, they do show a certain degree of variability. Given the enormous dataset the authors have at hand. Could a meta-analysis of such distance maps be constructed that would depict this inter-individual variability more clearly? Or is the variability due to random fluctuations of the model? How can biological and technical variability be distinguished?

The authors could discuss how further methodological improvement may advance this technology to the next level. Also, the limitations of using twin-networks could be elaborated upon.

Reviewer #3:

Remarks to the Author:

Toulany&al propose a deep-learning based method of zebrafish embryo staging. The authors recorded a very large number (~10 000) of movies of developing zebrafish embryos. Using a deep-learning neural network, they show that they can then automatically « stage » an embryo by calculating a similarity measure between an embryo image and their dataset. They show that this prediction becomes more

uncertain as developmental time progresses, and that abnormal embryos diverge from the general trend earlier than what the human eye would recognise. The authors show perturbed trajectories of similarity measures between unperturbed and perturbed embryos, following drug treatment. They also show that comparison of one embryo images to an earlier time points define broad « epochs » where embryo shapes are more comparable to each other than in between epochs.

Overall, this work is nicely done, clearly explained and relies on an interesting dataset of many developing embryos. It is nice to see a solid quantification of the so-far qualitative notion of « staging »; it is clear that this quantification is made possibly only by the use of deep learning and a very large dataset.

Beyond these positive points, I also find that the conclusions of the study are somewhat lacking in depth: we can see from the author analysis that perturbed embryos are different from unperturbed embryos; that some embryos do not develop normally; that there are recognisable broad stages of development based on embryo morphology; however it seems to me that none of these observations seems to go very far what can already be seen by eye. It is nice to see that this rigorous method recapitulates known observations, but how does it allow to go beyond them? For instance from the title referring to « tempo » I was expecting a more in-depth analysis of variability in developmental timing than what the manuscript provides.

Therefore, although the study introduces a new rigorous staging method, it is not clear to me at this point that the paper clearly shows this method can or will bring a significant change in the study of developmental dynamics.

Other comments:

- I think that the caption of Fig. 4a is not sufficiently descriptive, it is difficult to understand what the schematics mean.

Author Rebuttal to Initial comments

Response to editorial suggestions and reviewers' comments

Editorial suggestions

Your Article, "Uncovering developmental time and tempo using deep learning", has now been seen by 3 reviewers. As you will see from their comments below, although the reviewers find your work of considerable potential interest, they have raised a number of concerns. We are interested in the possibility of publishing your paper in Nature Methods, but would like to consider your response to these concerns before we reach a final decision on publication.

We therefore invite you to revise your manuscript to address these concerns. In particular, I recommend adding a strong biological demonstration to support the method. I also recommend adding discussion on whether this approach can be extended to analysis of other species.

Thank you. We were happy to see that the reviewers appreciated our work and found it of considerable potential interest. As detailed below, we have carefully addressed all of their comments.

We have also edited the manuscript according to your editorial guidance and recommendations. In particular, we have added a strong biological demonstration to support the method. Specifically, we have investigated temperature-dependent variability in the developmental tempo of zebrafish and medaka embryos. The results of this analysis are presented in the new Fig. 2, Supplementary Movies 2-3 and Supplementary Fig. 5-6. It is well known that poikilothermic animals adjust their development according to the temperature of their environment. Remarkably, medaka can thrive at an extreme temperature range from 10–40°C (Wittbrodt et al., 2002)! However, to our knowledge this phenomenon has not yet been quantitatively analyzed for medaka embryos. Classical physical biology theories (Arrhenius, 1889; van 't Hoff, 1884) predict an exponential relationship between reaction rates and ambient temperature. In striking agreement with these theories, we found that zebrafish and medaka developmental tempo predictably changed as a function of ambient temperature within species-specific bounds. This allowed us to calculate effective apparent activation energies of zebrafish and medaka growth. Interestingly, the activation energies of ~60/70 kJ/mol are much lower than those estimated for mammalian cells in tissue culture (Knapp & Huang, 2022). Our findings thus provide robust experimental support for the notion of an inverse relationship between apparent activation energies and the temperature ranges that support linear growth across different taxa (Knapp & Huang, 2022).

As suggested, the updated manuscript also includes a discussion on the extension of our approach to the analysis of other species. We expect that our methods will be widely applicable and useful to describe the development of uncharacterized species and to facilitate their use in studies of developmental dynamics and evolution.

Reviewer #1:

Remarks to the Author:

The authors present essentially a robust method to compute and quantify phenotypical differences between images of larvae. Application of this method to a huge number of images from developing zebrafish embryos, by comparing different stages, results in the definition of a development trajectory for normal embryos. This is then used to analyze variability between "normal" individuals during the first 24 hrs of development, to identify individuals that deviate from the normal trajectory, and to compare trajectories of normal embryos from that of embryos treated with inhibitors for BMP or Nodal signaling. Finally, the authors show that comparison of images from a single embryo, but at different stages, allows for definition of

developmental epochs and stages, which they also apply to less well characterized species such as medaka, stickleback, and even *C. elegans*.

This approach is certainly novel, it is convincing in the results it provides, and impressive in the scale of work presented.

Thank you for appreciating the novelty and value of our work.

Only minor points need to be addressed:

Minor points:

1) In line 62, the authors state that their dataset comprises 3 million images, in Fig. 1a they state 15 million: how comes? Could it be that the 15 million were required to obtain the 3 million "high quality" images used for model training? What happened to the remaining one's, were they ever used?

Thank you for pointing out the need for clarification. Indeed, 15 million images were acquired and later sorted automatically to exclude dead or malformed embryos as well as out-of-focus samples. This resulted in the final number of 3 million images. We have added an explanation to the *Data set cleaning* paragraph of the Materials and Methods section, and we have relabeled Fig. 1a to avoid confusion.

2) Lines 152-175: the authors clearly illustrate that defects due to depletion of specific signaling pathways will be detected by their developmental staging method, however without giving any information about the specific defect that causes the divergence from the normal staging trajectory. Possibly (presumably), treatment with another compound affecting BMP signaling would result in a trajectory indistinguishable from the one shown here.

Thank you very much for this idea! To test the hypothesis, we have now analyzed additional time-series of zebrafish embryos, in which BMP signaling had been perturbed in a different manner compared to the original LDN-193189 compound treatment. To make sure that we assessed *bona fide* BMP-loss-of-function, we used previously acquired time-lapse data for the homozygous BMP mutant *swirl* (<https://doi.org/10.48606/15>). The similarity profiles resulting from our analysis (Supplementary Fig. 7a) phenocopy the chemical inhibition (Fig. 4c) and result in very similar trajectories.

However, I also presume that very different phenotypes may occur at similar stages, thus yielding a similar deviation from the normal trajectory, but also from the BMP-inhibition one. Would it be useful to have a "developmental trajectory database" that could be used for new studies, on other compounds even in a different lab.

Thank you for pointing this out. We have performed new analyses with other inhibitors and different degrees of severity for modulated BMP signaling (Fig. 4 and Supplementary Fig. 7), which could be used as starting point for future reference in developing a developmental trajectory database. Our approach has the potential for creating such a developmental trajectory database encompassing different compound alterations and experimental conditions, but this would require generating a much larger collection of images that span a wide spectrum of resolutions, experimental scenarios and phenotypes beyond the scope of the current work. We have added this point as an outlook to Supplementary Note 3 of the manuscript (please also see our response to Reviewer #3 below).

Could another lab use the images from this work and run the comparison with its own images, or would all the images need to be recaptured (different setting, different conditions)?

We provide all of the data sets used in the manuscript as a permanent open resource (<https://doi.org/10.48606/15>). Our image data sets were taken with three different microscopes (please see the updated Materials and Methods section). For comparison with other microscopes, the models would need to be validated and potentially retrained for other conditions. However, we expect that the models we provide can be fine-tuned using small data sets, i.e. data that can be acquired in a single experiment.

3) Supplementary Fig. 6: I am not sure this representation is adding to the comprehension, given that Fig. 4b is already quite clear.

We agree and have removed the figure for clarity of presentation.

4) Supplementary Fig. 8c: there seems to be a mismatch between the stage timing in minutes indicated along the images and the stage timing in hours in the "autostage strip".

Thank you for pointing out this mistake. The timing in hours was based on a wrong frame rate of 2 min – rather than the correct 5 min. The values in *min* units below the images were correct, and we have now adjusted the values in *h* units according to the correct frame rate in the new Supplementary Fig. 10c (Supplementary Fig. 8c in the previous version of the manuscript).

5) How many images were used for the medaka, stickleback and *C. elegans* autostage determination? I finally found it in "Image acquisition", but it should be indicated maybe in the figure legends.

As suggested, we have specified the sample number directly in the figure legends for clarity of presentation.

A lower number of images may explain the plethora of autostages seen for stickleback, given the presumably also high variability that was also observed in zebrafish.

In addition to different developmental dynamics, the accuracy of autostaging could also be affected by the number of training images, embryo mobility, and the presence of unspecific structures such as lipid droplets present in medaka and stickleback embryos. In the updated manuscript, we have explored the effect of the number of training embryo images on the prediction of self-similarity matrices in zebrafish (Supplementary Fig. 9) and found that even a low number of training embryos produced robust results. The accuracy of autostaging can likely be further improved by stabilizing embryo positions using mounting media to reduce variability in the images. However, our preliminary experimental explorations in this direction using dechoriation, various mounting media, magnetic nanoparticle-based orientation, inhibition of embryo motility using bungarotoxin, n-heptanol and other anesthetics frequently resulted in developmental alterations, and are therefore currently not suitable for the analysis of normal developmental dynamics.

Marc Muller

Reviewer #2:

Remarks to the Author:

none This manuscript is a follow-up paper of a previously published study employing neural networks to phenotype embryogenesis in fish. The current work makes substantial advances and is conceptually different as it employs a twin-network architecture. This interesting methodology allows the authors to quantify morphological distances in an unbiased manner. It is applied to normal development and signaling perturbation and even extended to other species of fish and even *C. elegans*. In general, the paper is informative, presents a novel approach and appears scientifically sound and well documented. The biological findings confirm long known concepts and are rather unsurprising, thus this is rather a methodological paper that does not provide much novel biological insight. However, the potential to build upon this methodological framework is large and thus, I think it is of exceptionally high scientific interest to a larger audience.

Thank you for highlighting the value of our work.

I have two main questions:

Fig. 3 depicts how phenotypic differences after signaling perturbation can be detected in an unbiased manner. That is cool and has a multitude of potential applications.

Thank you for recognizing the potential of our method.

However, I didn't understand how robust in terms of sensitivity or specificity such a phenotyping method would be. Given that the perturbation is known, can the parameter space in which this is feasible be defined more clearly? I would love to get a feeling for how many embryos need to be treated and how penetrant the phenotype would have to be to be distinguishable from random noise, and how and where the detection would reach its limits. I know this depends on multiple, investigator specific parameters, threshold levels, ... So I don't expect an easy answer, but could the dataset the authors gathered serve as an example case to derive a clearer picture of how usable the twin-network phenotyping would be in an applied setting?

We agree with the reviewer and have now evaluated the performance of abnormality detection in a constrained yet wide range of phenotypes caused by the alteration of the major signaling pathways involved in vertebrate development. We have extended our initial analysis of phenotypes caused by *-BMP* and *-Nodal* (presented in the previous version of the manuscript) to those caused by loss-of-function of FGF (Fibroblast Growth Factor), PCP (Planar Cell Polarity), Shh (Sonic hedgehog) and Wnt signaling, as well as gain-of-function in RA (Retinoic Acid) signaling (Fig. 4 and Supplementary Movies 7-13). To determine the accuracy of our method, we performed the similarity comparison using a frame-by-frame detection approach to identify deviations from normal reference embryos. The detection method depended only on the significance level of the differences and the minimum fraction of temporal occurrence of the phenotype. As suggested by the reviewer, we evaluated how the accuracy of abnormality detection depends on the number of embryos for each of the analyzed phenotypes (Fig. 4j). This gives an estimate of the minimum number of embryos required to accurately identify different kinds of development defects. For 100% accuracy of detection, the number of required embryos ranged from 5 to 40 depending on the perturbation, which can be readily obtained for many applied settings.

To further analyze how penetrant the phenotype would have to be to be distinguishable from random noise, we systematically analyzed embryos treated with BMP inhibitors at different concentrations that elicit varying degrees of phenotype severity (Kishimoto et al., 1997). The performance of our Twin Network in detecting abnormalities indeed depended on phenotype severity: Strongly dorsalized *-BMP* phenotypes could be robustly identified, while the detection of very mildly dorsalized phenotypes with minor tail defects in comparison to wild-type embryos was reduced (Supplementary Fig. 7, Supplementary Movies 15-18).

Is there a way to quantify how robust this method would distinguish a phenotype without prior knowledge? What parameters are crucial in which ranges?

Thank you for these questions. We have designed our method with the goal that the detection of abnormalities does not rely on any specific prior knowledge. In contrast to classification approaches that depend on specific manual annotations – i.e. prior knowledge – our Twin Network approach only assumes that there is a deviation from normal developmental dynamics.

In the updated manuscript, we describe that robustness and performance of detection depend on the penetrance of the phenotype (Supplementary Fig. 7). Training our Twin Network requires about 100 embryos for reliable results, which can be readily obtained in one imaging session for many model systems. Phenotype detection can be tuned by two principal parameters: (i) The significance level for the differences, and (ii) the minimum temporal occurrence fraction of the phenotype. In addition, we found that a crucial parameter is the number of evaluated embryos. While strong phenotypes can be detected with just a few (4-5) embryos, weaker phenotypes require a couple of tens for accurate prediction (Supplementary Fig. 7f). In many applied settings, however, such embryo numbers can be readily obtained.

The distance matrices across developmental stages are impressive. It seems that Fig. 4 And those shown in Suppl Fig. 7 were each derived from a single embryo. Interestingly and unsurprisingly, they do show a certain degree of variability. Given the enormous dataset the authors have at hand. Could a meta-analysis of such distance maps be constructed that would depict this inter-individual variability more clearly?

Thank you for recognizing the power of the distance matrices and the insightful question.

To explore and distinguish technical from biological variability in self-similarity matrices, we have now trained ten models of TwinNet on a set of training images (61 embryos and 100,000 image triplets). All models were trained with the same embryo images and parameters, but the random image triplets and initial weights were different. Ten self-similarity matrices were then calculated for each embryo with one prediction per model.

With respect to technical variability, Supplementary Fig. 9c,e and Supplementary Fig. 9d,e show the element-wise average (i.e. ensemble matrix) and the standard deviation of the matrices for two zebrafish embryos, respectively. Even though there is some technical variability arising from different models and predictions, the main staging patterns are robustly identified.

To highlight the biological variability between embryos over technical variability, we calculated the average and standard deviation, respectively, of the ensemble matrices across 126 zebrafish embryos (Supplementary Fig. 9a,b). The major patterns in the stages are still visible, but clearly more diffuse compared to single embryos.

The data describing these findings have been incorporated into the updated manuscript, and the Materials and Methods sections has been updated accordingly.

The authors could discuss how further methodological improvement may advance this technology to the next level. Also, the limitations of using twin-networks could be elaborated upon.

Thank you for this excellent suggestion. Given space constraints in the main text, we have added the following discussion as Supplementary Note 3 to the manuscript:

Our approach provides a generalized framework for comparing microscopy images, which opens the possibility not only to generate specialized models tailored to specific applications, but also to produce more general and robust models not restricted to a single problem. Future methodological improvements could involve generating a comprehensive collection of images that span a wide spectrum of resolutions, experimental scenarios, or even encompass images captured using diverse microscopy techniques from various species. Methodological enhancement employing Generative Adversarial Networks (GANs) to create expansive data sets could be also useful when experimental data is scarce. A limitation of such an approach is that the use of Twin Networks could potentially introduce biases if the training data is not appropriately selected. However, if properly utilized, this has the potential of a developmental trajectory database encompassing different species and experimental conditions, which could be a game-changer for evolutionary developmental biology applications. Such a database could facilitate cross-species similarity comparisons, shedding light on the underlying principles of developmental processes and evolution.

Reviewer #3:

Remarks to the Author:

Toulany&al propose a deep-learning based method of zebrafish embryo staging. The authors recorded a very large number (~10 000) of movies of developing zebrafish embryos. Using a deep-learning neural network, they show that they can then automatically « stage » an embryo by calculating a similarity measure between an embryo image and their dataset. They show that this prediction becomes more uncertain as developmental time progresses, and that abnormal embryos diverge from the general trend earlier than what the human eye would recognise. The authors show perturbed trajectories of similarity measures between unperturbed and perturbed embryos, following drug treatment. They also show that comparison of one embryo images to an earlier time points define broad « epochs » where embryo shapes are more comparable to each other than in between epochs.

Overall, this work is nicely done, clearly explained and relies on an interesting dataset of many developing embryos. It is nice to see a solid quantification of the so-far qualitative notion of « staging »; it is clear that this quantification is made possibly only by the use of deep learning and a very large dataset.

Thank you very much. Our method provides an unbiased and standardized way to stage, compare and synchronize embryos, and we anticipate that these capabilities will make our approach useful in a wide range of applications.

Beyond these positive points, I also find that the conclusions of the study are somewhat lacking in depth: we can see from the author analysis that perturbed embryos are different from unperturbed embryos; that some embryos do not develop normally; that there are recognisable broad stages of development based on embryo morphology; however it seems to me that none of these observations seems to go very far what can already be seen by eye.

We demonstrate that our method can be used to detect phenotypic variability within a population. This not only provided new insights into the non-linearity of development and major embryonic branching points, but also allowed us to detect embryos that deviated from normal developmental trajectories. We predict that these capabilities will pave the way for new studies to analyze the molecular basis of developmental robustness. Our approach is independent of specific annotations and can therefore detect any alteration from normal developmental dynamics.

Our approach can also be used to automatically generate atlases of the major developmental epochs in diverse species – from zebrafish to medaka and *C. elegans*. Indeed, given the diverse morphology of fish embryos, even experienced zebrafish researchers struggle with detecting corresponding stages in other evolutionarily distant fish species such as medaka and stickleback. We therefore expect that our modular approach will be useful to describe the development of uncharacterized species and to facilitate their use in studies of development and evolution.

It is nice to see that this rigorous method recapitulates known observations, but how does it allow to go beyond them? For instance from the title referring to « tempo » I was expecting a more in-depth analysis of variability in developmental timing than what the manuscript provides.

Thank you for this comment. To showcase the potential of our method in generating new insights into developmental dynamics, we investigated temperature-dependent variability in the developmental tempo of zebrafish and medaka embryos. The results of this analysis are presented in the new Fig. 2, Supplementary Movies 2-3 and Supplementary Fig. 5-6. It is well known that poikilothermic animals adjust their development according to the temperature of their environment. Remarkably, medaka can thrive at an extreme temperature range from 10–40°C (Wittbrodt et al., 2002)! However, to our knowledge this phenomenon has not yet been quantitatively analyzed for medaka embryos. Classical physical biology theories (Arrhenius, 1889; van 't Hoff, 1884) predict an exponential relationship between reaction rates and ambient temperature. In striking agreement with these theories, we found that zebrafish and medaka developmental tempo predictably changed as a function of ambient temperature within species-specific bounds. This allowed us to calculate effective apparent activation energies of zebrafish and medaka growth. Interestingly, the activation energies of ~60/70 kJ/mol are much lower than those estimated for mammalian cells in tissue culture (Knapp & Huang, 2022). Our findings thus provide robust experimental support for the notion of an inverse relationship between apparent activation energies and the temperature ranges that support linear growth across different taxa (Knapp & Huang, 2022).

In addition to these novel biological insights, we would like to highlight the technical advantages of our method compared to other recent approaches (Crapse et al., 2021). Since our method is automated and independent of manual annotations, we were able to perform this analysis in weeks compared to years, and on hundreds compared to dozens of embryos each at different temperatures. Most importantly, our method is independent of a finite number of pre-specified stages, enabling the analysis of developmental tempo as a continuum in time. These advantages and our findings underscore the importance of automated techniques in comprehending intricate biological phenomena, opening new possibilities for further research and application in diverse biological systems.

Therefore, although the study introduces a new rigorous staging method, it is not clear to me

at this point that the paper clearly shows this method can or will bring a significant change in the study of developmental dynamics.

We are convinced that our methods will bring at least four significant changes in the study of developmental dynamics. First, our approach provides an unbiased and standardized way to stage, compare and synchronize embryos, and we anticipate that these capabilities will make our approach useful in a wide range of applications. For example, our physicist colleagues have told us that staging is notoriously hard for them, and our standardized approach will facilitate many of their valuable developmental biophysics studies. Second, our approach allows the rapid analysis of large numbers of embryos in a high-throughput manner, which would otherwise be lengthy or prohibitively manually labor-intensive. This effective feature is highlighted by our novel insights into temperature-dependent developmental dynamics (Fig. 2): Our experiments and Twin Network analyses were executed with hundreds of embryos and within weeks, whereas similar approaches based on other current technologies would be limited to smaller sample sizes, require months or even years, and may be prone to human error. Third, our approach can be used to detect phenotypic variability within a population, and we predict that these capabilities will pave the way for new studies to analyze the molecular basis of developmental robustness with cutting-edge omics approaches. Fourth, our approach can be used to automatically generate atlases of the major developmental epochs in diverse species – from zebrafish to *C. elegans*. Indeed, we expect that our modular approach will be useful to describe the development of uncharacterized species and to facilitate their use in studies of development and evolution.

Other comments:

- I think that the caption of Fig. 4a is not sufficiently descriptive, it is difficult to understand what the schematics mean.

Thank you for pointing this out. We have rewritten the Fig. 5 legend (Fig. 4 in the previous version of our manuscript) for clarity, and we hope that the schematic is now accessible to a broad readership.

References

- Arrhenius, S. A. (1889). Über die Reaktionsgeschwindigkeit bei der Inversion von Rohrzucker durch Säuren. *Z. Phys. Chem.*, 4, 226–248. <https://doi.org/doi:10.1515/zpch-1889-0416>
- Crapse, J., Pappireddi, N., Gupta, M., Shvartsman, S. Y., Wieschaus, E., & Wuhr, M. (2021). Evaluating the Arrhenius equation for developmental processes. *Mol Syst Biol*, 17(8), e9895. <https://doi.org/10.15252/msb.20209895>
- Kishimoto, Y., Lee, K. H., Zon, L., Hammerschmidt, M., & Schulte-Merker, S. (1997). The molecular nature of zebrafish swirl: BMP2 function is essential during early dorsoventral patterning. *Development*, 124(22), 4457–4466. <https://doi.org/10.1242/dev.124.22.4457>
- Knapp, B. D., & Huang, K. C. (2022). The effects of temperature on cellular physiology. *Annu Rev Biophys*, 51, 499–526. <https://doi.org/10.1146/annurev-biophys-112221-074832>
- van 't Hoff, J. H. (1884). *Etudes de dynamique chimique*. Frederik Müller.
- Wittbrodt, J., Shima, A., & Schartl, M. (2002). Medaka—a model organism from the far East. *Nat Rev Genet*, 3(1), 53–64. <https://doi.org/10.1038/nrg704>

Decision Letter, first revision:

Dear Patrick,

Thank you for submitting your revised manuscript "Uncovering developmental time and tempo using deep learning" (NMETH-A52047A). It has now been seen by the original referees and their comments are below. The reviewers find that the paper has improved in revision, and therefore we'll be happy in principle to publish it in Nature Methods, pending minor revisions to satisfy the referees' final requests and to comply with our editorial and formatting guidelines.

TRANSPARENT PEER REVIEW

Nature Methods offers a transparent peer review option for new original research manuscripts submitted from 17th February 2021. We encourage increased transparency in peer review by publishing the reviewer comments, author rebuttal letters and editorial decision letters if the authors agree. Such peer review material is made available as a supplementary peer review file. Please state in the cover letter 'I wish to participate in transparent peer review' if you want to opt in, or 'I do not wish to participate in transparent peer review' if you don't. Failure to state your preference will result in delays in accepting your manuscript for publication.

ORCID

Sincerely,
Madhura

Madhura Mukhopadhyay, PhD
Senior Editor
Nature Methods

Reviewer #1 (Remarks to the Author):

As previously stated, the authors present a method to define developmental trajectories using images from zebrafish embryos. The authors extended their original submission by adding the analysis of zebrafish and medakas at different temperatures, as well as in the presence of different signalling inhibitors or mutations. These new data add to the validity and applicability of the method, a more broader perspective is also included. I consider that my comments, and those of the other reviewers were accurately addressed.

Reviewer #2 (Remarks to the Author):

I find the additional data included in the revised manuscript impressive and insightful. The authors all of my concerns in full. I suggest to publish this paper in Nat Methods.

Reviewer #3 (Remarks to the Author):

In their revision, Tulany & al have answered my comments and notably have added a study of temperature-dependence of development in Fig. 2 which strongly strengthens the manuscript. Therefore I recommend publication of this work.

Author Rebuttal, first revision:

Response to reviewers' comments

Reviewer #1:

Remarks to the Author:

As previously stated, the authors present a method to define developmental trajectories using images from zebrafish embryos. The authors extended their original submission by adding the analysis of zebrafish and medakas at different temperatures, as well as in the presence of different signalling inhibitors or mutations. These new data add to the validity and applicability of the method, a more broader perspective is also included. I consider that my comments, and those of the other reviewers were accurately addressed.

Thank you for your thorough review and feedback. We appreciate the time you took to evaluate our work and are glad to hear that you find the additions and modifications satisfactory. The input from you and the other reviewers has been invaluable in improving the quality and scope of our study.

Reviewer #2:

Remarks to the Author:

I find the additional data included in the revised manuscript impressive and insightful. The authors all of my concerns in full. I suggest to publish this paper in *Nat Methods*.

Thank you for your positive feedback and endorsement for publication in *Nature Methods*. We are grateful for your thorough review and are pleased to hear that our revisions have addressed your concerns satisfactorily. Your insights and suggestions have greatly contributed to the improvement of our manuscript.

Reviewer #3:

Remarks to the Author:

In their revision, Tulany & al have answered my comments and notably have added a study of temperature-dependence of development in Fig. 2 which strongly strengthens the manuscript. Therefore I recommend publication of this work.

Thank you for your constructive feedback and recommendation for publication. We appreciate your recognition of the improvements made, particularly regarding the addition of the temperature-dependence study in Fig. 2. Your insights have been instrumental in enhancing the quality of our manuscript.

Final Decision Letter:

Dear Patrick,

I am pleased to inform you that your Article, "Uncovering developmental time and tempo using deep learning", has now been accepted for publication in Nature Methods. Your paper is tentatively scheduled for publication in our December print issue, and will be published online prior to that. The received and accepted dates will be 19 Mar 2023 and 15 Oct 2023. This note is intended to let you know what to expect from us over the next month or so, and to let you know where to address any further questions.

Over the next few weeks, your paper will be copyedited to ensure that it conforms to Nature Methods style. Once your paper is typeset, you will receive an email with a link to choose the appropriate publishing options for your paper and our Author Services team will be in touch regarding any additional information that may be required.

You will receive a link to your electronic proof via email with a request to make any corrections within 48 hours. If, when you receive your proof, you cannot meet this deadline, please inform us at rjsproduction@springernature.com immediately.

Please note that *Nature Methods* is a Transformative Journal (TJ). Authors may publish their research with us through the traditional subscription access route or make their paper immediately open access through payment of an article-processing charge (APC). Authors will not be required to make a final decision about access to their article until it has been accepted. [Find out more about Transformative Journals](https://www.springernature.com/gp/open-research/transformative-journals)

Authors may need to take specific actions to achieve [compliance](https://www.springernature.com/gp/open-research/funding/policy-compliance-faqs) with funder and institutional open access mandates. If your research is supported by a funder that requires immediate open access (e.g. according to [Plan S principles](https://www.springernature.com/gp/open-research/plan-s-compliance)) then you should select the gold OA route, and we will direct you to the compliant route where possible. For authors selecting the subscription publication route, the journal's standard licensing terms will need

to be accepted, including [self-archiving policies](https://www.springernature.com/gp/open-research/policies/journal-policies). Those licensing terms will supersede any other terms that the author or any third party may assert apply to any version of the manuscript.

Your paper will now be copyedited to ensure that it conforms to Nature Methods style. Once proofs are generated, they will be sent to you electronically and you will be asked to send a corrected version within 24 hours. It is extremely important that you let us know now whether you will be difficult to contact over the next month. If this is the case, we ask that you send us the contact information (email, phone and fax) of someone who will be able to check the proofs and deal with any last-minute problems.

If, when you receive your proof, you cannot meet the deadline, please inform us at rjsproduction@springernature.com immediately.

Once your manuscript is typeset and you have completed the appropriate grant of rights, you will receive a link to your electronic proof via email with a request to make any corrections within 48 hours. If, when you receive your proof, you cannot meet this deadline, please inform us at rjsproduction@springernature.com immediately.

Once your paper has been scheduled for online publication, the Nature press office will be in touch to confirm the details.

Once your paper has been scheduled for online publication, the Nature press office will be in touch to confirm the details.

Content is published online weekly on Mondays and Thursdays, and the embargo is set at 16:00 London time (GMT)/11:00 am US Eastern time (EST) on the day of publication. If you need to know the exact publication date or when the news embargo will be lifted, please contact our press office after you have submitted your proof corrections. Now is the time to inform your Public Relations or Press Office about your paper, as they might be interested in promoting its publication. This will allow them time to prepare an accurate and satisfactory press release. Include your manuscript tracking number NMETH-A52047B and the name of the journal, which they will need when they contact our office.

About one week before your paper is published online, we shall be distributing a press release to news organizations worldwide, which may include details of your work. We are happy for your institution or funding agency to prepare its own press release, but it must mention the embargo date and Nature Methods. Our Press Office will contact you closer to the time of publication, but if you or your Press Office have any inquiries in the meantime, please contact press@nature.com.

Nature Portfolio journals [encourage authors to share their step-by-step experimental protocols](https://www.nature.com/nature-research/editorial-policies/reporting-standards#protocols) on a protocol sharing platform of their choice. Nature Portfolio 's Protocol Exchange is a free-to-use and open resource for protocols; protocols deposited in Protocol Exchange are citable and can be linked from the published article. More details can found at www.nature.com/protocolexchange/about.

Best regards,
Madhura

Madhura Mukhopadhyay, PhD
Senior Editor
Nature Methods